# Tropical forest carbon sequestration accelerated by nitrogen

Wenguang Tang [1,2], Jefferson S. Hall [3], Oliver L. Phillips [1], Roel J. W. Brienen [1], S. Joseph Wright [4], Michelle Y. Wong [4,5,6], Lars O. Hedin [7], Michiel van Breugel [4,8], Joseph B. Yavitt [9], Phillip M. Hannam [5] & Sarah A. Batterman [1,4,5] ✉

Understanding forest carbon sequestration is crucial for predicting and managing the carbon cycle, yet we lack evidence for whether, when and how the carbon sink in tropical forests recovering from land use change is nutrient limited. Here we show how the tropical forest recovery rate responds to experimental nutrient manipulation over a secondary succession gradient in a naturally recovering Central American landscape. Nutrient limitation of aboveground biomass accumulation shifts from strong nitrogen limitation in young forests to no evidence of nitrogen or phosphorus limitation in older secondary or mature forests. Nitrogen addition increases aboveground biomass accumulation by 95% in recently abandoned pasture and 48% in 10-year-old forests. Conversely, we observe no influence of nitrogen on older forests and no evidence of phosphorus limitation at any stage. If our findings of nitrogen limitation extend to young tropical forests globally, nitrogen could prevent the sequestration of 0.69 (0.47-0.84) Gt $CO_2$ each year.

Tropical forests play a key role in the terrestrial carbon sink by sequestering atmospheric carbon dioxide ($CO_2$) and thus slowing the rate of global climate change[1–3]. Mature tropical forests harbor substantial quantities of biomass carbon, and young secondary tropical forests (*i.e.*, those naturally regenerating from disturbance) offer a large potential carbon sink as a natural climate solution[1,4], owing to their increasing importance in modern tropical landscapes and their potential for high rates of carbon accumulation relative to mature forests[1,5–7].

While phosphorus is often thought to limit carbon sequestration by tropical forests on highly weathered lowland soils[8,9] (but see refs. 10–13), we know little about how nutrients and carbon interact throughout the course of tropical forest recovery from disturbance. One possibility is that nutrient limitation of tree growth is independent of forest successional age. Alternatively, nutrient limitation may

depend on forest age, with a shift from nitrogen to phosphorus limitation due to differences in nutrient losses, sources and accumulation rates for each nutrient following disturbance. Disturbances like blowdowns, logging, fire and agriculture cause losses of soil nitrogen through leaching and gaseous emissions, exacerbating nitrogen limitation[14–17]. As secondary succession proceeds, symbiotic nitrogen fixation can build up ecosystem nitrogen over time[18,19], which, in turn, can alleviate nitrogen limitation and shift the forest to limitation by another resource, usually assumed to be phosphorus. Soil phosphorus may or may not decrease following disturbance – and can even increase after forest cutting and burning[20,21] – and is unlikely to increase with secondary succession since there are few inputs of new phosphorus to tropical forests over such short time scales[15,22]. Thus, tropical forests may – in theory – shift from nitrogen to phosphorus limitation over secondary succession.

[1]School of Geography, University of Leeds, Leeds, UK. [2]School of Geographical and Earth Sciences, University of Glasgow, Glasgow, UK. [3]ForestGEO, Smithsonian Tropical Research Institute, Ancón, Panamá, Panama. [4]Smithsonian Tropical Research Institute, Apartado 0843–03092, Balboa, Panama. [5]Cary Institute of Ecosystem Studies, Millbrook, NY, USA. [6]Department of Ecology and Evolutionary Biology, Yale University, New Haven, CT, USA. [7]Department of Ecology and Evolutionary Biology, Princeton University, Princeton, NJ, USA. [8]Department of Geography, National University of Singapore, Singapore, Singapore. [9]Department of Natural Resources and the Environment, Cornell University, Ithaca, NY, USA. ✉e-mail: battermans@caryinstitute.org

Evidence that soil nutrients are likely to limit the tropical forest carbon sink and that there could be shifts over succession derives from multiple sources. Nutrient addition experiments find that nutrients can limit tropical tree growth[23–28], with nutrient limitation stronger in young forests[24]. Several experiments suggest nitrogen limitation of young forests (refs. including[25,29]) and that nitrogen or phosphorus limitation can occur in mature forests (refs. including[25,26,28,30–32]). A meta-analysis across 44 experiments found that both nitrogen and phosphorus limit young but not mature tropical forests[24]. None of these experiments included both young and mature forests in the same area, limiting inference about shifts in limitation. Direct observations of tropical forest nitrogen cycles generally show decreasing symbiotic nitrogen fixation and increasing nitrogen losses to the atmosphere and groundwater as forests mature, suggesting nitrogen limitation in young forests that declines with forest age[15,18,19,33] (but see refs. 21,34–36). Biogeochemical model predictions provide indirect evidence for a similar nitrogen limitation decline and shift to phosphorus limitation[14,37], and dynamic global vegetation models indicate the future land carbon sink will be constrained by nitrogen and/or phosphorus[38–41]. Thus, although some evidence suggests tropical forest nutrient limitation shifts from nitrogen to phosphorus over secondary succession, experimental evidence shows the potential for both types of limitation, and evaluating nutrient limitations across disparate experimental sites could be confounded by other variables. It therefore remains unclear whether and how nutrient limitation evolves over secondary succession.

To address this knowledge gap, we established a landscape-scale experiment in lowland moist tropical forests of Panama. The experiment was designed to directly test whether nitrogen, phosphorus or the two nutrients combined affect stand-scale aboveground biomass and rates of biomass net change, gain and loss, and whether the effects change over forest successional age. Because our 76 (0.1–0.16 ha) plots

were located across a ~16 km² landscape, our experiment was able to control for metacommunity species pool, land use history, soils and climate while avoiding pseudo-replication. We distributed our plots across a secondary succession gradient that included newly regenerating forests ("0-year-old forests"), two middle stages ("10-" and "30-year-old forests") and a mature forest occurring on land that has had limited human disturbance ("600-year-old forests"; Supplementary Fig. 1, "Methods"). In our sites, soil total nitrogen ranges between 0.28–0.35%, and soil total phosphorus ranges between 246 to 286 mg P kg⁻¹ in the top 30 cm and does not change over forest succession (Supplementary Tables 1–3). The early successional forest sites were previously clear-cut for cattle pasture, with no burning since initial deforestation 30–40 years ago. We monitored forest changes over four censuses (four years in secondary forests and 21 years in the mature forest). In total, we monitored the responses of 88,843 trees to nutrient manipulation. A resolution to tropical nutrient limitation, even at one site, would aid the implementation and management of tropical reforestation to increase carbon sequestration as a natural climate solution. Our experiment also offers an opportunity to validate dynamic global vegetation models (DGVMs) and improve predictions of the future of tropical forests.

## Results and discussion

Stand-scale aboveground biomass (determined by individual biomass and stem density) recovered rapidly following the abandonment of cattle pasture, independent of the addition of nutrients (black line in Fig. 1). By 14 years, our forests contained 25% of the aboveground biomass of mature forests, and by 30 years, they had accumulated half the aboveground biomass of mature forests. These biomass recovery rates are similar to mean rates for secondary forests across the American tropics[5]. With added nutrients, however, the aboveground biomass in our forests recovered even more rapidly. Across

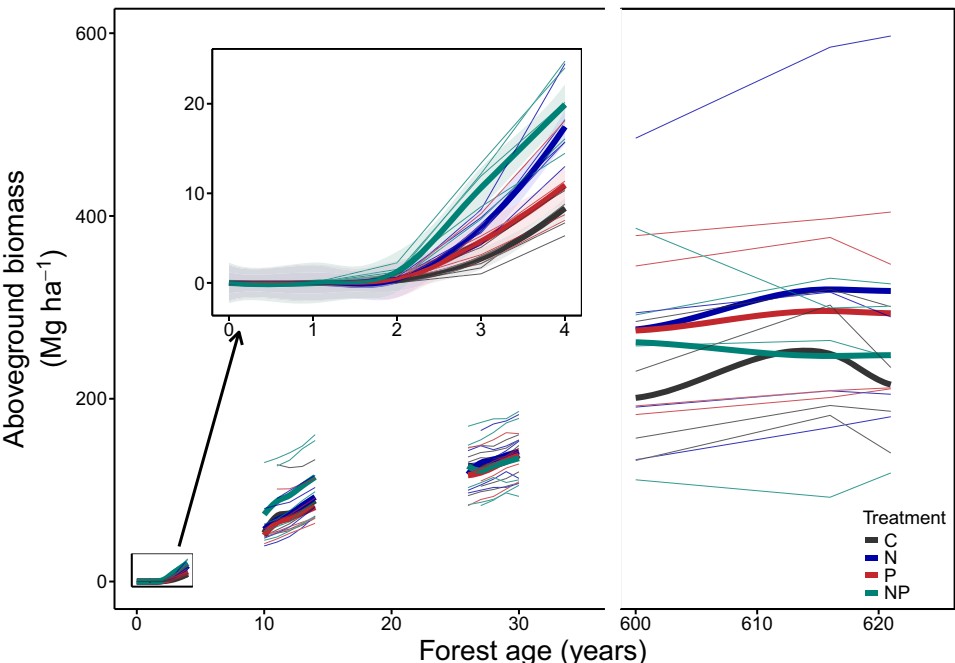

**Fig. 1 | Aboveground biomass (Mg ha⁻¹) and its response to nutrient additions over tropical forest secondary succession (years post disturbance).** Different treatments are represented by different colors. Black lines represent the control treatment (C, no nutrient addition); blue lines, the nitrogen addition treatment (N); red lines, the phosphorus addition treatment (P); green lines, the nitrogen plus phosphorus treatment (NP). Thin lines represent the aboveground biomass in each plot across censuses, and thick lines represent the mean of each treatment, fitted with a loess fit. The lines in the inset panel, which zoom in on the aboveground biomass in the 0-year-old forest, include 95% confidence intervals fitted using the 'loess' method. Five replicates were present per forest age and treatment in secondary forests, except for the first census of both 10- and 30-year-old forests, where there were only four replicates. The mature forests have four replicate plots per census.

**Table 1 | Statistical results from linear mixed effects models of the effects of nitrogen, phosphorus, forest age and their interactions on forest aboveground biomass net change, gain and loss**

| Response variable | Model | Variable | Sum Sq | p-value | $R^2_{marginal}$ | $R^2_{conditional}$ |
|---|---|---|---|---|---|---|
| Biomass net change | N*P*Age + random(Block) | **N** | **18.1** | **0.026** | 0.74 | 0.76 |
| | | P | 0.7 | 1.000 | | |
| | | **Age** | **229.5** | **<0.001** | | |
| | | N*P | 4.3 | 0.853 | | |
| | | **N*Age** | **39.6** | **0.003** | | |
| | | P*Age | 12.9 | 0.559 | | |
| | | N*P*Age | 6.9 | 1.000 | | |
| Biomass gain | N*P*Age + random(Block) | **N** | **33.3** | **<0.001** | 0.76 | 0.80 |
| | | P | 0.6 | 1.000 | | |
| | | **Age** | **120.6** | **<0.001** | | |
| | | N*P | 1.5 | 1.000 | | |
| | | **N*Age** | **31.6** | **<0.001** | | |
| | | P*Age | 9.8 | 0.3534 | | |
| | | N*P*Age | 2.2 | 1.000 | | |
| Biomass loss | N*P*Age + random(Block) | N | 2.0 | 0.238 | 0.86 | 0.87 |
| | | **P** | **3.4** | **0.040** | | |
| | | **Age** | **107.5** | **<0.001** | | |
| | | N*P | 1.3 | 0.556 | | |
| | | **N*Age** | **7.5** | **0.007** | | |
| | | **P*Age** | **18.5** | **<0.001** | | |
| | | N*P*Age | 0.6 | 1.000 | | |

The p-values reflect the Holm adjustment to address the problem of multiple comparisons. Block accounts for spatial variation, with the four treatments (control, nitrogen (N), phosphorus (P), nitrogen plus phosphorus (NP)) replicated in four (mature forest) or five (secondary forests) spatial groups across the landscape and is treated as a random effect variable in the model. Bold denotes significant effects at p < 0.05.

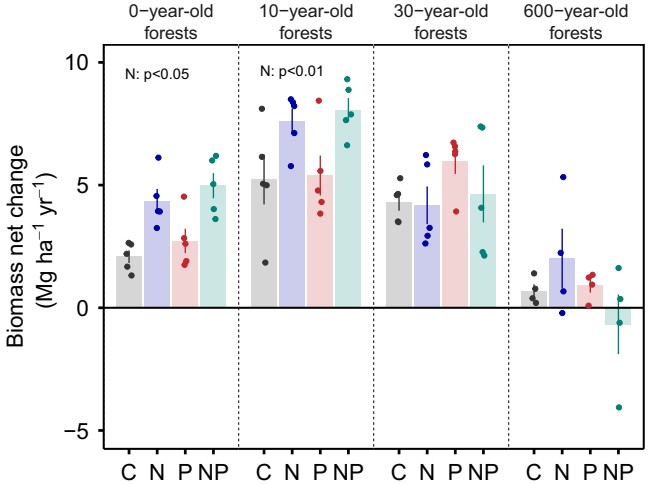

**Fig. 2 | The effect of nitrogen and phosphorus on aboveground biomass net change over tropical forest secondary succession.** Nutrient treatments are displayed on the x-axis, including control (C), nitrogen (N), phosphorus (P) and nitrogen plus phosphorus (NP). Points represent plot-level mean net biomass change for the fertilization period (2015–2019 for the young forests, 1997–2018 for the mature forest), with each bar representing the mean ± standard error of the mean (s.e.m.) across plots (n = 5 and 4 plots for young secondary and mature forests, respectively). Significant effects of nitrogen addition on biomass net change are present in both 0- and 10-year-old forests (p < 0.05). See Table 1, Supplementary Table 4 for full statistical results.

treatments, nitrogen had the largest and only statistically significant effect on aboveground biomass, with four years of nitrogen addition increasing biomass by 95% and 33% in the 0- and 10-year-old forests, respectively (green and blue lines versus black and red lines in Fig. 1).

Nutrient limitation of net aboveground biomass accumulation (measured as the change in plot biomass from one census to the next) shifted over secondary succession. The 0- and 10-year-old tropical forests were strongly limited by nitrogen, while the 30- and 600-year-old forests were not (Table 1, Fig. 2). The interaction between nitrogen addition and forest age was statistically significant for net aboveground biomass change (Table 1). Nitrogen addition caused 95% and 48% increases in net aboveground biomass accumulation rates for the 0- and 10-year-old forests, respectively (Fig. 2, Supplementary Table 4 and Supplementary Figs. 2, 3). Nitrogen limitation in the control plots, therefore, prevented the sequestration of ~4.1 tons $CO_2$ ha$^{-1}$ year$^{-1}$ over the first decade of tropical forest recovery ("Methods"). In contrast, there was no nitrogen effect on net aboveground biomass change in the 30-year-old secondary forest or after twenty years of sustained nutrient addition in the mature forest (Fig. 2, Supplementary Table 4 and Supplementary Figs. 4, 5). Thus, we conclude that nitrogen limits the net accumulation of aboveground biomass in our young successional forests but not in forests 30 years and older.

Next, we sought to understand the mechanism by which nutrient addition influences the dynamical balance between the growth and mortality of aboveground biomass across our forest successional gradient. To do this, we decomposed net aboveground biomass change into two demographic components: First, the gain in aboveground biomass caused by the growth of trees that survived between census intervals, plus any trees that recruited into the plots between censuses. And second, the loss of aboveground biomass caused by the mortality of trees between censuses.

We found that, similar to net aboveground biomass change, the addition of nitrogen interacted with forest age (Fig. 3a, Table 1), triggering a 99% increase in aboveground biomass gain in 0-year-old forests (Fig. 3a, Supplementary Table 4) and a 23% gain in 10-year-old forests. Also similar to our results for net aboveground biomass change, we observed no statistically significant nitrogen effect on aboveground biomass gain in forests 30 years and older (Fig. 3a,

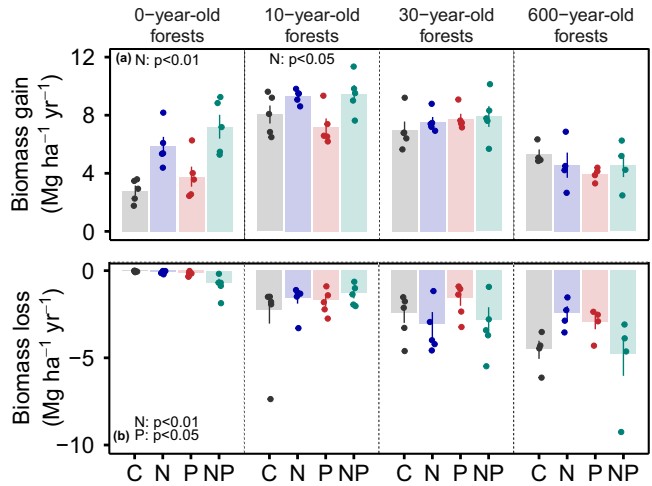

**Fig. 3 | The effect of nitrogen and phosphorus on aboveground biomass gain and loss over tropical forest secondary succession.** Nutrient treatments are displayed on the x-axis, including control (C), nitrogen (N), phosphorus (P) and nitrogen plus phosphorus (NP). Points represent plot-level mean aboveground biomass gain and loss for the fertilization period (2015–2019 for the young forests, 1997–2018 for the mature forest), with each bar representing the mean ± standard error of the mean (s.e.m.) across plots ($n$ = 5 and 4 plots for young secondary and mature forests, respectively). Significant effects of nutrient addition on aboveground biomass gain and loss are present in both 0- and 10-year-old forests ($p < 0.05$). See Table 1, Supplementary Table 4 for full statistical results and Supplementary Fig. 2 for a magnification of aboveground biomass loss in the 0-year-old forests.

Supplementary Table 4 and Supplementary Figs. 4, 5). Nitrogen addition also influenced aboveground biomass loss, with substantial biomass loss due to mortality in the 0-year-old forests (Table 1, Supplementary Table 4), likely caused by self-thinning, resulting from increased competition between the rapidly growing plants[42]. We found no statistically significant effect, however, of nitrogen on aboveground biomass loss in forests 10 years and older (Fig. 3b, Supplementary Table 4 and Supplementary Figs. 3–5).

We found little evidence of phosphorus limitation on stand-scale aboveground biomass dynamics in our tropical moist forests, either across forest ages or in interaction with forest age (Table 1). Furthermore, there was no significant effect of phosphorus on aboveground biomass net change, gain or loss, with the exception of aboveground biomass loss in the 0-year-old forests (Fig. 3, Table 1 and Supplementary Table 4). Note these findings are counter to expectations from biogeochemical theory[8,9] (but see ref. 37) and consistent with analyses of individual tree growth in our mature forest[11].

Taken together, our findings provide direct support for the hypothesis that nutrient limitation on aboveground biomass shifts over secondary succession in tropical forests, from nitrogen limitation early in succession to limitation by another resource later in succession. Our findings emerge from a landscape-scale experiment that spans a secondary succession gradient from abandoned pasture to mature forests and captures the effect of nutrients on stand-scale net aboveground biomass accumulation. Although they derive from one site, these findings advance our understanding of the role that nitrogen and phosphorus can have in tropical forests.

Two findings from our study stand out. First, the strength of nitrogen limitation in our young forests is high, double that of secondary forests up to a decade older (i.e., ref. 29 and our 10-year-old forests). Second, nitrogen limitation decreases as forests age, and the resource limiting later successional forests – or any forest age – was not phosphorus, as expected by prevailing biogeochemical theory[8]. Because of the propensity for nitrogen – but not rock-derived phosphorus – to be lost from ecosystems following disturbance, the

possibility that rock-derived nutrients may even increase following forest cutting and burning[21], the high nitrogen requirements of recovering vegetation, and the previous evidence suggesting nitrogen limitation in other young tropical forests, we expect that nitrogen limitation in young secondary tropical moist forests may be common. There may, nonetheless, be heterogeneity across tropical forests if nitrogen is especially important in locations where phosphorus is in sufficient supply, but less so where soil nitrogen stocks are substantial or phosphorus is particularly low (e.g., ref. 34). Heterogeneity in nitrogen versus phosphorus availability across tropical forests could explain why different experiments indicate nitrogen or phosphorus limitation in young forests[24]. Additionally, young tropical forests could be co-limited by nitrogen and phosphorus, whereby the addition of phosphorus allows the stimulation of nitrogen fixation, which would alleviate nitrogen limitation.

Our findings expand previous lines of evidence of nitrogen limitation in young tropical forests. First, a meta-analysis of 31 nutrient addition experiments, 24 of which were at the individual tree scale and eight of which were in secondary forests, has shown that tree growth may respond to soil nitrogen in tropical forests, and that this occurs most strongly in secondary forests[24]. Second, one of two fertilization experiments that tested nitrogen limitation on stand-scale net aboveground biomass change found that nitrogen addition led to 67% more aboveground biomass in 6-year-old secondary tropical forests[29] (Supplementary Table 5). The second experiment found no evidence of nitrogen limitation on net aboveground biomass change in 60-year-old secondary forests[43]. Findings from these experiments are consistent with a decline in nitrogen limitation over succession. No experiment with stand-scale findings that we are aware of has found nitrogen limitation of aboveground biomass in tropical forests at later stages of primary succession or has tested a shift in limitation from secondary to mature tropical forests. Third, also at the stand scale, three studies using observational indicators of the nitrogen cycle over secondary succession found conservative nitrogen cycles in early secondary succession, consistent with nitrogen limitation, but not in later succession[15,19,21]. Finally, mechanistic models have predicted that nitrogen may limit aboveground biomass accumulation in young secondary forests and that this declines as succession proceeds[14,17,37]. Our landscape-scale ecosystem experiment across a secondary succession gradient allows us to advance these previous findings by directly testing for nutrient limitation on ecosystem-scale aboveground biomass carbon accumulation for a range of forest ages while controlling for soil properties, metacommunity species pool, climate and other factors.

Our lack of support for the widely held hypothesis of phosphorus limitation on tropical forest net aboveground biomass accumulation does not appear to be caused by exceptionally high levels of soil phosphorus across our forests. Soil in our sites – at 246 to 286 mg P kg⁻¹ (Supplementary Tables 2, 3) – are within the range of total phosphorus observed across the Amazon (25–1000 mg P kg⁻¹)[44] and fall between the relatively phosphorus-poor eastern (< 200 mg P kg⁻¹) and phosphorus-rich western Amazon (> 300 mg P kg⁻¹)[45]. About one-quarter of Amazonian forest areas are located on soils that have more phosphorus than our sites (Supplementary Table 3)[44]. Thus, we conclude that our findings of a lack of phosphorus limitation on stand-scale aboveground biomass accumulation may apply to some other tropical forests worldwide. Future work should examine the degree to which these patterns are consistent in other tropical forests, including African and Asian tropical forests.

Recent work suggests that our lack of phosphorus limitation in younger tropical forests could have emerged from phosphorus enrichment following slash-and-burn agricultural practices[21]. However, it seems unlikely that this is the reason for lack of phosphorus limitation in our young forests because 1) our sites have had limited

slash-and-burn practices that occurred more than 30–40 years ago, and 2) our soil data do not show increased pH or phosphorus in the young compared to old forests (Supplementary Table 1), as would be expected if slash-and-burn practices changed the phosphorus cycle and the nature of nutrient limitation in tropical forests. Nonetheless, it is possible that, within a site, our young forest soils have higher phosphorus than they had originally because of the initial deforestation and burning. We do not have the pre-deforestation soil nutrient measures for our forest plots to evaluate this.

Another possibility for why we did not observe evidence of phosphorus limitation is that it may not be expressed in wood productivity, but instead, affects the productivity of other tissues. Indeed, production of litter[46] and plant reproductive structures[47] appears to increase with phosphorus addition in our mature forests. Phosphorus addition triggered responses in fine roots and litter – but not wood production – in a low-phosphorus site in Amazonia[10]. While it is plausible that limitations on stem wood production may be expressed on much longer timescales, substantial stand-scale biomass growth responses have been documented after as little as 1.5 years of fertilization in a Hawaiian mature tropical forest[32]. In contrast, our mature forest plots showed no evidence of phosphorus limitation on wood productivity even after two decades of continuous addition.

The timescale over which tree growth is flexible and able to respond to nutrients is critical for whether we can definitively conclude that a lack of growth response to fertilizer addition is due to a lack of nutrient limitation on the system. Our experiment shows that young forest trees are sufficiently flexible to respond in growth to nutrient addition over a matter of years. Whether or not mature forest species can do this is less clear. Nearby mature forests with similar climate and metacommunity have up to two times greater aboveground woody production than our mature forest[48,49], suggesting that higher growth rates of our mature forests are biophysically possible. Thus, the lack of growth response to nutrients in our experiment could be because low-phosphorus-adapted trees in our mature forests lack the ability to respond to phosphorus addition over short timescales[23]. Changes in mature forest tree growth may instead occur over decades to centuries via shifts in tree community composition as tree species turn over in response to changing nutrient conditions[50].

It is also possible that phosphorus limitation on stand-scale wood productivity in tropical forests is less common – and heterogeneity in nutrient limitation more common – than previously thought. Theory on biogeochemical cycling in tropical forests has primarily been developed from mature forests, and in Hawaii[51], where soil phosphorus levels are more than double (680 mg kg$^{-1}$)[52] and tree biodiversity (~20 species in total[53]) orders of magnitude lower than tropical forests in Central and South America (~166 mg P kg$^{-1}$, Supplementary Table 3, and ~16,000 species[54]). In fact, direct evidence for phosphorus limitation on stand-scale aboveground biomass carbon accumulation in the tropics is limited (summarized in Supplementary Table 5), potentially because trees have evolved nutrient acquisition and use strategies to tolerate and even thrive on low phosphorus soils. We have observed some evidence that this occurs in our forests because we found increased root phosphatase activity in our 30-year-old and mature forests relative to the young forests, and that phosphorus activity is downregulated in response to added phosphorus[55,56]. These strategies may be particularly beneficial in older forests where additional nutrient requirements may be smaller than in younger forests. Furthermore, evidence suggests that total phosphorus may not be a good indicator of phosphorus availability in a tropical forest system and that, although measurable pools of available phosphorus are often relatively small[44], phosphorus can be mobilized from unavailable pools rapidly through biotic mineralization of organic phosphorus and desorption of inorganic phosphorus[57]. This rapid mineralization and desorption provides a substantial supply of new inorganic phosphorus that can be quickly taken up by trees. Nonetheless, evidence suggests

that phosphorus may limit aboveground biomass in some tropical forests (e.g., ref. 32). In other areas, however, increasing evidence indicates that other soil resources like potassium[11,27], calcium[21] or water[58,59] can limit mature tropical forests. Our findings point to the need to understand the interactive effects of nitrogen and other soil resources, with nitrogen playing an important role in early succession and other (or no) soil resources limiting forests in later succession.

This study provides experimental evidence of the interaction between the nitrogen and carbon cycles, and, combined with previous observations[18] and model predictions[17], demonstrates that a carbon-nitrogen feedback mechanism can occur in tropical moist forests. Following disturbance, rapid carbon accumulation and strong nitrogen limitation stimulate symbiotic nitrogen fixation, which increases nitrogen availability. Subsequently, the strength of nitrogen limitation declines and carbon recovery accelerates as succession proceeds. Alleviation of nitrogen limitation leads to the downregulation of fixation in later succession. Although model predictions suggest that symbiotic nitrogen fixation can double the rate of carbon accumulation in young secondary forests[17], we found that our young forests were still nitrogen-limited even with the presence of symbiotic nitrogen fixation. Nitrogen fixation may not completely overcome nitrogen limitation immediately after disturbance[60] because of dispersal limitation of legume trees or bacteria[17] or constraints by another resource like phosphorus or molybdenum[61] or by herbivory[62], none of which were considered in the model[17].

Our findings, albeit from only one site, combined with previous experimental and observational findings, can help inform how we represent tropical forests in regional and global carbon models. Next-generation Earth system, climate change and ecosystem models should reflect our emerging understanding of the heterogeneity of nutrient limitation in tropical forests, with at least some young tropical forests being nitrogen-limited and some older forests not being limited by phosphorus. Models should capture the carbon-nitrogen-phosphorus dynamics observed in our experiment and suggested at other sites (as discussed in the introduction) over the course of forest succession that may result in a change in nutrient limitation under global change. Our work suggests that forests recovering from disturbance may be especially prone to nutrient limitation. Given increasing rates of disturbance and $CO_2$ supply, nutrient limitation will likely be an increasingly critical feature of tropical carbon dynamics and sinks[23,24,63,64]. Mature forests may also tip to nitrogen limitation as small-scale disturbances like forest gaps increase[65], as suggested by previous findings that symbiotic nitrogen fixation upregulates in mature forest gaps[66]. Next-generation dynamic global vegetation models should capture the dynamics of coupled biogeochemical cycles that likely exist in some tropical forests beyond our site.

Our experimental findings, combined with previous field[18] and model[17] findings on symbiotic nitrogen fixation from the same sites, have implications for how we understand and manage tropical forests for natural climate solutions. Avoiding deforestation of mature tropical forests should always be prioritized, but our finding of the nutrient enhancement of secondary forest carbon accumulation is important as policymakers evaluate where and how to restore forests to maximize carbon sequestration.

We estimate that nitrogen limitation may prevent the sequestration of up to an additional 0.84 Gt $CO_2$ year$^{-1}$ between 2020 and 2050 (mean 0.69 Gt $CO_2$ year$^{-1}$, range 0.47 to 0.84 Gt $CO_2$ year$^{-1}$) if tropical reforestation is used as a natural climate solution (Supplementary Information 2)[67]. This amount depends on the degree to which our findings of nitrogen limitation on young forests extend to tropical forests globally – a substantial uncertainty. It also depends on the strength of nitrogen limitation, the cost of carbon and if economic measures were taken to alleviate nitrogen limitation.

Three enhanced reforestation practices for avoiding nitrogen limitation could increase carbon sequestration. First, include native

tree species that fix nitrogen in tropical reforestation, including as supplemental plantings in natural regeneration. This practice offers a natural way to accelerate secondary forest carbon sequestration, while avoiding excess nitrogen and nitrous oxide emission in mature forests since nitrogen-fixing trees downregulate fixation in mature forests[18,66]. It may not, however, alleviate all nitrogen limitation. Second, reforestation could be prioritized in areas with higher nitrogen deposition. Nitrogen deposition is projected to increase in tropical regions[68] and enhance the forest carbon sink[69]. Beyond the carbon benefit, regrowing forests would avoid soil acidification, depletion of other nutrients and emissions of other greenhouse gasses such as nitrous oxide by absorbing localized excess nitrogen[70]. Third, reforestation could be focused on higher nutrient soils. This practice risks putting reforestation in competition with agriculture and may face ethical questions and political limitations at scale, though carbon pricing incentives could induce land-use switching. The implications for global food supply require further analysis, including optimization of the food system according to productivity, nutrient and carbon considerations. Leakage may be avoided with emerging jurisdictional approaches to carbon auditing[71]. While directly fertilizing forests might meet nutrient requirements more rapidly, its feasibility is questionable given the cost and risks of increasing fertilizer-related energy consumption and emissions of the powerful greenhouse gas nitrous oxide (Supplementary Information 2). Combining all three management practices may be ideal, but any one of them would lower the cost of realizing reforestation potential and increase the $CO_2$ abatement available via reforestation, enabling a faster carbon sink and buffering time to achieve 1.5 or 2 °C[67]. Although this approach would only advance carbon sequestration by a few decades, the next few decades are crucial to give the global economy time to decarbonize. Fundamentally, efforts to understand and manage the future carbon sink in tropical forests must account for the governing role of nutrients.

## Methods

### Research sites

A factorial nitrogen and phosphorus fertilization experiment was established at Agua Salud (9°13'N, 79°47'W, 330 meters above sea level) and Gigante (9°06'31"N, 79°50'37"W, 60 meters above sea level) in the Republic of Panama (Supplementary Fig. 1). These two sites lie within the Panama Canal Watershed and include forests at different stages of succession (from young secondary to mature forest). Both forest sites are classified as lowland tropical moist forests, receiving similar annual precipitation (~2700 mm) with a dry season (contributing ~10% of total rainfall) from December to April, having a similar mean annual temperature (26 °C), and sharing a similar diverse metacommunity of tree species[18,23,72,73]. Soils across all forests are highly weathered with generally low plant-available soil nutrient concentrations[18,74]. Gigante contains deep soils on relatively flat terrain and has been classified as Oxisols (Typic Kandiudox), and Agua Salud soils are classified as Inceptisols (Typic Dystrudept; five of six soil pits) and Oxisols (Typic Haplodux; one of six pits). The soil physical and chemical properties of Agua Salud and Gigante forests can be found in the Supplementary Table 1. The soil classification reports can be found in ref. 75.

In Agua Salud, the landscape consists of cattle pastures and cultivated fields, fallows, plantations and different age secondary forests which recovered naturally following cattle ranching and small-scale clear-cutting[18,73]. Topography in this area varies, consisting of narrow streams and steep but short slopes[73]. In Gigante, which is a part of the Barro Colorado Nature Monument, the land is covered by a protected mature forest on relatively flat terrain[76].

### Experimental design

The Gigante fertilization experiment (named '600-year-old forest') was established in a mature forest of at least 600 years post-human disturbance, as determined with charcoal dating in 1997[77]. It consists of four nutrient addition treatments (control, nitrogen, phosphorus and nitrogen plus phosphorus) with each replicated four times (1 forest age × 4 treatments × 4 replicates)[23]. The area of each plot is 0.16 ha (40 × 40 m). The Agua Salud Project fertilization experiment consists of experimental plots at three different successional stages: very young secondary forests established immediately after the abandonment of cattle pastures (named "0-year-old forests") and two middleaged secondary forests (named "10-year-old forests" and "30-year-old forests"). The experiment began in 2015 with plot establishment of four replicates per treatment for the 10- and 30-year-old forests and the establishment of four replicate former pasture plots that were clear of trees as 0-year-old forests. In 2016, we established a fifth replicate set of plots for all forest ages and began fertilization treatments with the same nutrient addition treatments as the mature forests. In both sites, within each replicate, we blocked the control, nitrogen, phosphorus and nitrogen plus phosphorus plots within sites on the landscape to minimize the effects of small-scale variations in climate, soils and surrounding forest tree diversity and seed source (Supplementary Fig. 1 in Supplementary Information 1). Plots are located on the landscape to eliminate the possibility of the flow of nutrients from one plot to another based on slope, and the minimum distance between plots is 40 m. The area of fertilized plots is 0.16 ha (40 × 40 m), and the control plots are 0.1 ha (20 × 50 m) since they do not need a fertilization buffer. In every Agua Salud fertilization plot, trees are monitored only within the inner 0.1 ha, leaving a buffer zone on four sides that was fertilized.

Fertilizer was added as coated urea ($(NH_2)_2CO$) and triple superphosphate ($Ca(H_2PO_4)_2 \cdot H_2O$) in nitrogen and phosphorus-treated plots, respectively. Annual doses were 125 kg N ha$^{-1}$ yr$^{-1}$ and 50 kg P ha$^{-1}$ yr$^{-1}$, and fertilizers were added by hand in four equal doses at regular intervals beginning approximately two weeks after the beginning of regular, heavy rains (approximately 15–30 May, 1–15 July, 1–15 September and 15–30 October)[11].

We also measured the annual rainfall when the Agua Salud fertilization experiment was established (see the annual rainfall variation in the Supplementary Fig. 6). We found that changes in rainfall do not affect biomass accumulation and its dynamics.

### Forest inventory

We monitored all 76 plots since the start of the nutrient fertilization (i.e., 2015 in Agua Salud forests and 1997 in Gigante forest). All freestanding woody plants (trees, palms and lianas – hereon referred to as trees since 75% of plants were trees) within the plots were identified, but the monitoring protocols differed slightly between the two sites. In Agua Salud, in the center 0.1 ha of plots, all stems of trees and palms with diameters ≥ 5 cm and all lianas with diameter ≥ 1 cm were measured, as well as 50% of all tree and palm stems with diameters between 1 and 5 cm. All trees of the same size cutoffs were measured in the control plots. In Gigante, trees with a diameter ≥ 10 cm were measured in the whole plot, and trees with a diameter between 1 and 10 cm were measured in the central 20 × 30 m subplot. We use the nested design to capture the dynamics of small trees. For the large trees, diameters were measured above any buttresses or other deformities of the lower trunk[23,73]. All diameters were measured at 1.3 meters height.

In Agua Salud, plots were censused prior to fertilization in 2015, with the exception of the plots established in 2016. Plots were then censused yearly after fertilization from 2016 to 2019. In Gigante, plots were censused every five years from 1997 to 2018; however, we focused on the census prior to fertilization in 1997 and the censuses after fertilization from 2003 to 2018. We divided our analysis into two census intervals to capture (1) the most recent forest dynamics that would be comparable to the climatic conditions at the Agua Salud experiment (2013–2018) and (2) the forest characteristics prior to fertilization and the subsequent post-fertilization dynamics (the 1997 to 2013 censuses).

## Biomass at the plot scale

We first estimated the aboveground biomass (kg stem⁻¹) of all recorded stems in Agua Salud and Gigante plots. We applied different allometric functions to estimate the aboveground biomass of each stem of each tree, liana and palm. For trees, we estimated the aboveground biomass of each stem using the following allometric function[78]:

$$AGB = \exp[-1.803 - 0.976E + 0.976\ln(WD) + 2.673\ln(DBH) - 0.0299[\ln(DBH)^2].$$

where $AGB$ represents aboveground biomass (kg stem⁻¹), $E$ is the local climatic index[79], and $WD$ is wood density (g cm⁻³). $DBH$ represents the diameter at 1.3 meter (cm). The climate index, $E = 0.05645985$, near our study site, represents the effect of the environment on tree height allometry[79]. Species-specific wood density (in g cm⁻³) was estimated from the most common species in Agua Salud and Gigante (S. Joseph Wright unpublished data and ref. 79).

For lianas, the aboveground biomass of each stem was calculated using a liana-specific allometric equation[80,81]:

$$AGB = \exp[-0.999 + 2.682*\ln(DBH)].$$

For palms, we calculated the aboveground biomass using a palm-specific allometric equation[79,82]:

$$AGB = 0.0417565*(DBH)^{2.7483}.$$

We summed the aboveground biomass of all stems one centimeter and above in each plot and scaled to one hectare to get the plot scale per hectare biomass (Mg ha⁻¹, Fig. 1). To account for differences in the methods of inventorying all stems five centimeters and above and stems one to five centimeters in half of the plot at Agua Salud, we cloned the aboveground biomass of stems one to five centimeters to double the aboveground biomass of stems in that size class. At Gigante, where we inventoried all stems greater than ten centimeters and stems one to ten centimeters in 37.5% of the plot area, we summed the aboveground biomass of stems one to ten centimeters and multiplied by 2.667.

## Forest dynamics at the plot scale

We calculated the mean annual net change of aboveground biomass for each plot and census interval (between 2015 and 2016, 2016 and 2017, 2017 and 2018, and 2018 and 2019 for Agua Salud plots, and between 1997 and 2013, and 2013 and 2018 for Gigante plots; Mg ha⁻¹ year⁻¹; Fig. 2, Supplementary Figs. 2–5).

We then calculated plot scale aboveground biomass gains and losses for each census interval in Agua Salud plots (the same intervals as described above), and for longer intervals in the Gigante plots (between 1997 and 2003, between 2003 and 2008, between 2008 and 2013, and between 2013 and 2018; Mg ha⁻¹ year⁻¹; Fig. 3, Supplementary Figs. 2–5). For Gigante, we then took the mean of the dynamics between 1997 and 2003, between 2003 and 2008, and between 2008 and 2013 to get the dynamics for the interval 1997 to 2013. We then took the mean dynamics of these census intervals to represent the full fertilization period. Growth was calculated as the gains of the trees recorded in the first census year that survived until the next census, divided by the time between the two censuses. Recruitment was calculated as the total aboveground biomass gain of trees that were recorded in the later census but not in the previous, divided by the time between the two censuses. Mortality was calculated as the aboveground biomass loss because of the loss of trees recorded in the initial census by the second census, divided by the period between the two censuses. Due to potential systematic error of overestimating growth and underestimating recruitment because of the cutoff of minimum tree size, we combined growth and recruitment into one measure of biomass gain and considered mortality as biomass loss.

Finally, for forest ages where we found significant effects of nutrients, we converted the net aboveground biomass change due to nutrients into $CO_2$ equivalent. We took the difference in aboveground biomass sequestered between the forests with and without the added nutrient into $CO_2$ by multiplying by 0.47 (the carbon:biomass ratio[83]) and 3.66 (the number of $CO_2$ molecules per carbon). We then took the mean of the values at 0 and 10 years to calculate the mean $CO_2$ accumulation due to nutrient addition over the first ten years of forest recovery (tons $CO_2$ ha⁻¹ year⁻¹).

## Statistical analysis

All analyses were performed in R (4.0.2)[84]. We used linear mixed-effects models ('lmer' function in "lme4" package) to test for the effects of nutrient addition and forest age on aboveground biomass net change, aboveground biomass gain and aboveground biomass loss at the plot scale as a mean across censuses. The mixed-effects models included nitrogen, phosphorus, forest age and their interactions, with block – the grouping of treatments across the landscape – as a random effect. In each model, all fixed effects were treated as factors. We log-transformed aboveground biomass loss to meet the model assumptions. For all models, we used the "anova" function for results and checked the residual and Q-Q plots to evaluate model assumptions. We evaluated whether annual rainfall influenced forest dynamics at Agua Salud; however, the effects were minimal (Supplementary Fig. 6).

We use the 'Holm' adjustment to avoid the problem of multiple comparisons[85]. When we found a significant interaction of a nutrient and age, we used contrasts (the 'emmeans' function from the 'emmeans' package) with the 'Holm' adjustment to determine the specific age at which the nutrient was limiting. We considered an effect to be significant if $p < 0.05$. The marginal and conditional $R^2$ were obtained for the models with the 'r2' function from the 'r2mlm' package.

## Data availability

The data generated for this study have been deposited in the Figshare database under accession code CC BY 4.0: https://doi.org/10.25390/caryinstitute.25892776.v1.

## Code availability

R Code generated for this study have been deposited in the Figshare database under accession code CC BY 4.0: https://doi.org/10.25390/caryinstitute.25892776.v1.

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

## Acknowledgments

This work is a contribution of the Agua Salud Project and the Gigante Fertilization experiment. The Agua Salud Project is a collaboration between the Smithsonian Tropical Research Institute (STRI), the Panama Canal Authority (ACP), and the Ministry of the Environment of Panama (MiAmbiente), is part of the Smithsonian Institution Forest Global Earth Observatory (ForestGEO) and is funded by the Heising-Simons Foundation, the Carbon Mitigation Initiative at Princeton University, the Leverhulme Trust, the United Kingdom Natural Environment Research Council Council (NE/M019497/1, NE/N012542/1), the British Council 275556724 with additional support from Stanley Motta, Frank and Kristin Levinson, the Hoch family, and the U Trust. The Gigante fertilization experiment was supported by the Andrew W. Mellon Foundation and Scholarly Studies Program of the Smithsonian Institution. W.G.T. was supported by a Chinese Scholarship Council-University of Leeds joint scholarship and Priestley Center for Climate Futures, the University of Leeds and the University of Glasgow. M.W. was supported by the Cary Institute Lang Assael Family Innovation Fund, the Millbrook Garden Club and Yale University. M.v.B. was supported by Yale-NUS and Singapore's Ministry of Education (IG19_SG113). We thank Dayana Agudo, Sebastian Bernal, Mario Bailon, Johanna Balbuena, Will Barker, Carlos Diaz, Guillermo Fernandez, Omar Hernández, Roderick Martinez, Victorino Montenegro, Andrew Nottingham, Miguel Núñez, Adriana Tapia, Julio Rodriguez, and Bonnie Waring.

## Author contributions

S.A.B. and J.S.H. established the Agua Salud fertilization experiment with input from L.O.H. and M.v.B. S.J.W. and J.B.Y. designed and established the Gigante fertilization experiment. S.A.B. and W.G.T. designed this research with input from O.L.P., R.J.W.B., J.S.H., S.A.B., S.J.W., and W.G.T. collected data. W.G.T. analyzed the data with S.A.B., O.L.P., R.J.W.B., M.W., P.M.H., and S.A.B. did the economic analysis. W.G.T. and S.A.B. wrote the manuscript with O.L.P. and R.J.W.B. All coauthors discussed the results and provided feedback on the manuscript.

## Competing interests

The authors declare no competing interests.
