## [Peer Review file · Nature Communications]

Tropical forest carbon sequestration accelerated by nitrogen

Corresponding Author: Dr Sarah Batterman

Version 0:

Reviewer comments:

Reviewer #1

(Remarks to the Author)

I reviewed an earlier version of this manuscript submitted to Nature. Compared to the earlier version there are several things that have improved substantially. Several earlier, somewhat exaggerated claims have disappeared and language usage is now much more concise compared to the earlier version. The authors also better acknowledge and discuss earlier work on N limitation in tropical forests which also improves the clarity on what is new and noteworthy in the current study.

When considering the critical question how well the soils in Panama represent soils under tropical forests, it was interesting to learn that the authors now acknowledge that both sites (Gigante and Agua Salud) also have less weathered soils than Oxisols. The authors however argue that the total P values are more important than the classification and I agree with that.

In their answer to my earlier questions regarding the total P values of 600 mg/kg the authors explain that the high number came from the paper by (Mirabello et al., 2013) which apparently overestimated the P in the 'hot' fractions. I looked up this paper and indeed they suggest that silica interference in phosphate detection by molybdate colorimetry may have led to an overestimation. Fortunately, total P was also analyzed in the (Koehler et al., 2009) paper, which was mentioned by the authors in their rebuttal. Koehler et al. (2009) was conducted in the Gigante Experiment and they used pressure digestion in concentrated HNO₃ followed by analysis of the digests using inductively coupled plasma-atomic emission spectrometer and report them in Table 1 of their paper. This total P method does not use colorimetry for analyses and thus does not suffer from potential silicon interference. The numbers reported by Koehler et al, 2009 for the control plots of the Gigante experiment are 550 mg kg⁻¹ in the top 0.05m of mineral soil and still as much as 400 mg kg⁻¹ in the 0.05 – 0.5m depth interval. If we take the lower value of 400 mg kg⁻¹, this would still put the Gigante site in the top 10% values of total P in the Quesada et al., 2010 paper. So even when I accept that the 600ppm reported by Mirabello et al., 2013 were overestimated because of methodological problems, the total P values of Gigante are more comparable to the top 10% of Amazonian forests. And as I pointed out in my review of the earlier version, this was exactly what I would expect of soils developed from basaltic parent material in tectonically active regions such as Panama. However, the real question for this manuscript is whether the total P levels are so important to extrapolate the result to other parts of the tropics and below I will argue, why I think that is not the case.

In their rebuttal the authors also write that it may be possible that the younger secondary forests could have a stronger nutrient (P) limitation compared to mature forests due to the rapid relative growth rates of young trees. In the manuscript, theoretical biogeochemical models are also mentioned as the basis for this expectation (l. 44 – l.49). Furthermore, they write counter to expectations from biogeochemical theory, they found little evidence of P limitation in the experiment (L. 122). Although this may be true, I think that the authors miss to appreciate the potential importance of land use history in this case (and in general with secondary forests). A strong P limitation in young secondary forest is unlikely because the secondary forest grows on areas that were previously cleared which probably involved slash-and-burn procedures and potentially agricultural practices. The results of slash and burn is that an important part of the rock derived-nutrients, including P, end up in the soil and the soil pH increases, effects that can last for decades. A higher pH leads to higher P availability and a higher base saturation which means that it is quite unlikely that rock-derived nutrients are limiting in the earlier successional stages. Something that was more or less shown for P by (Wong et al., 2024). To my knowledge, the Walker and Syers model considers natural systems without considering previous land use. However, taking the previous land use into account will go a long way to explain at least some of the findings in your study and I think that at the moment this is not sufficiently done.

You discuss the possibility that P limitation is not observed in wood productivity (l 177 and further), you discuss that it is possibly less common than previously thought (L 189 and further), but you never discuss that forest clearing and burning leads to substantial increases in rock-derived nutrients and their availability, which I think is currently missing. You should also not forget that the data from the (Quesada et al., 2010) paper were collected in old-growth forest, where presumably no slash and burn has been taken place in the last centuries. This is presently ignored in the manuscript when you compare your soils with soil of the Amazon Basin, and I really think it is misleading and causes a lot of confusion (at least it took me two reviews to realize what was wrong). I expect that young secondary forests and abandoned agricultural areas will have higher values in total and available nutrients because of former slash-and-burn activities, and that should be highlighted in your discussion. It should not be too difficult to back this up with some studies. BTW, thank you for providing the full soil analysis for the Agua Salud project. Your soil chemical analyses show pH-water values around 5 which is a pH value when soil P is relatively well available. Also, base saturation is regularly above 50% in the top soil which may be the result of ashes from earlier forest clearing and also indicates that these are not very nutrient poor soils.

I am still not convinced about your implications, in your rebuttal you write that you especially focus on tree planting when suggesting to increase the abundance of N fixing species. I have no doubt that there are some examples of secondary forests with little N fixing trees. However, the (Gei et al., 2018) paper, which was mentioned in the rebuttal, also shows that the majority of secondary forests have a relative basal area of 10% or (much) more. I am not sure whether adding more N-fixing legumes would make a big difference and as far as I know the vast majority of secondary forests is the result of spontaneous regeneration and succession and not from planting.

In summary, I think the study presented here definitely gives new, noteworthy and exciting insights in to shifts in nitrogen and phosphorus limitations over a secondary succession gradient in soils that are at the higher end of total phosphorus content of soils under (secondary) forests, and as such is worth publishing. I think your claim that your soils are close to the median value for tropical forests soils globally is simply not true and frankly, I don't think it is important for your manuscript. What is important for your message is that in secondary forests, N limitation is decreasing over time, and in my interpretation, it is unlikely to be replaced in the first decades of a secondary forest by P limitation (not even on heavily weathered Oxisols in the Amazon) because of the higher and more available rock-derived nutrient stocks, including soil P, which is a remnant of former slash and burn activities.

References

- Gei, M., et al., 2018. Legume abundance along successional and rainfall gradients in Neotropical forests. *Nat. Ecol. Evol.* 2, 1104–1111. <https://doi.org/10.1038/s41559-018-0559-6>
- Koehler, B., et al. 2009. Immediate and long-term nitrogen oxide emissions from tropical forest soils exposed to elevated nitrogen input. *Glob. Change Biol.* 15, 2049–2066. <https://doi.org/10.1111/j.1365-2486.2008.01826.x>
- Mirabello, M.J., et al. 2013. Soil phosphorus responses to chronic nutrient fertilisation and seasonal drought in a humid lowland forest, Panama. *Soil Res.* 51, 215. <https://doi.org/10.1071/sr12188>
- Quesada, C.A., et al., 2010. Variations in chemical and physical properties of Amazon forest soils in relation to their genesis. *Biogeosciences* 7, 1515–1541. <https://doi.org/10.5194/bg-7-1515-2010>
- Wong, M.Y., et al. 2024. Trees adjust nutrient acquisition strategies across tropical forest secondary succession. *New Phytol.* 243, 132–144. <https://doi.org/10.1111/nph.19812>

(Remarks on code availability)

Reviewer #2

(Remarks to the Author)

This is the second time I have reviewed this paper (albeit for a different journal). So I am copying out my previous review below since it addresses many of the stock questions asked by NCOMMS.

The authors now present the carbon numbers with appropriate nuance. They have also directly addressed my concern about nitrogen being confounded with age at the beginning. And the methods are now clearly documented, and are sound. I have no additional critiques beyond the two minor grammatical catches below.

Line 39: “increasing predominance in modern tropical landscapes” would be better than “increasing importance” since the latter is normative

Line 194: in LATIN America? Hawaii is also USA.

The paper by Tang et al examines impacts of fertilization on a network of tropical forest plots within Panama. They provide supplemental nitrogen and phosphorus to forests that vary in age (~0, 10, 30, and 600 years old) and find that the nitrogen addition greatly augments growth in the young forests only and that – despite standard biogeochemical theory – woody biomass accumulation does not appear to be limited by phosphorus.

The paper is very well written. I have no editorial critiques for clarity, and this never happens when I review a manuscript. As I was reading, multiple questions arose in my mind but they addressed these later in the manuscript. For example, they find

that phosphorus is not limiting which is a surprising result, but then provide a very nice discussion to explore potential mechanisms underlying their results. I don't think the paper's findings are enough to upend theory, but it certainly suggests that more work is needed to determine the universality of phosphorus limitation in the tropics.

I was concerned that the interaction between nitrogen and forest age was confounded by nitrogen conditions in the soil (i.e., more nitrogen intact forest soils), but they provide baseline soil fertility conditions in a supplemental information (Extended Data 2) that shows this is not the case. I suggest making that point explicitly within the main manuscript.

I was also concerned that the applied take home from the paper would be that people should apply fertilizer to young regrowing forests to augment growth and carbon removal. But the paper very clearly states that this is not recommended (I agree!) and provide alternative options suggest as prioritizing regrowth in places with N deposition.

However, the largest issue is reproducibility. The methods are non-existent. The paragraph located lines 68-78 seems written to make the experiment sound larger than it actually is. They say there are 76 plots, but I think this includes both the treatment and the control so actually the number of experimental units is smaller. Sample size per treatment appears to be restricted to the figure legends. They say that these cover 88,843 trees, but those are not independent replicates. They report that they monitored these plots for up to 21 years and then mention in the parentheses that secondary forests have been monitored for four, but in fact the main results of the paper come from those secondary forests. How large are the plots? They say they cover 16 km², but what is the range of sizes for the individual plots? Where are they located (i.e., can the geolocations be made available)? How much fertilizer did they add? What dosage, what frequency? How did they quantify biomass (i.e., which allometric equations were used)? This is all easily remedied by including a clear methods section – in the supplement if no space is available in the main text.

My only other critique is that their carbon numbers require some nuance. I appreciate that they estimated mitigation potential at a cost-constrained rather than unconstrained level. However, their number remains quite theoretical. It is highly unlikely/impossible to achieve 1.1 GtCO₂/yr because there are only so many locations with N deposition or higher fertility soils and only so much N that can be provided by nitrogen fixing species. Adding fertilizer to relieve nutrient limitations would produce upstream and in-field emissions that would offset the climate benefit of enhanced forest growth. I suggest the co-authors nuance that result more appropriately. For example, in the abstract instead of “could sequester up to an additional 1.1 GtCO₂/yr” it might say “N limitations prevent up to 1.1 GtCO₂/yr”. In other words, that 1.1 GtCO₂/yr shouldn't be presented as an opportunity.

Optional suggestions:

(1) Table 1 could be supplemental

(2) Figure 1 should be checked for color blind friendliness. And would a log-scale for the y-axis better illustrate the differences among treatments (as an alternative to the inset approach).

(Remarks on code availability)

Code and metadata are appropriate.

Reviewer #3

(Remarks to the Author)

This study evaluates the effects of fertilization on aboveground biomass accumulation in early, mid and late successional forests in Panama. The work itself is incredibly ambitious, and the results are very robust and of interest. This study was a major undertaking. The manuscript is also beautifully written, and the authors certainly present some intriguing ideas. I read an earlier draft of this manuscript, and there are substantial improvements. I appreciate the broader acknowledgement prior work, and the more developed discussion of alternate explanations for the lack of response to N and P fertilization in mature forests. That said, many of my prior reservations hold.

1. Abstract: I appreciate that in the discussion, the authors compare this work to forests in the Amazon, but I am still extremely hesitant about generalizing findings from a single forest site in Panama to the tropics as a whole, especially given the wide range of tropical forest types (i.e., seasonal, evergreen, flooded, etc) and that the neotropics may not reflect what we would see in Asian forests or other less studied systems.

2. Regarding the following statement –

“To date, we are aware of no direct ecosystem experiment that investigates how nutrient limitation on stand-scale aboveground carbon dynamics evolves over the course of a complete tropical forest secondary succession gradient, from recently abandoned pastures to mature forests.”

And this matters why? Being the first, or only, in and of itself isn't that interesting, but something that helps fill a specific knowledge gap, is. Please elaborate on why we as a scientific community should care about having this additional detail, and if the experiment just supports what is already generally accepted amongst tropical forest ecologists (i.e., early successional forests are N limited), is this really transforming our understanding? Definitely this is a really well-designed study, and the findings are very robust, but it just supports what others have shown - although maybe not as well. The authors suggest that some element other than P is what is really limiting mature tropical forests, but what do they think it is?. Also -- why would we expect mature trees to respond in the same way as fast growing early successional trees that are working to access the forest canopy?

3. The following statement “These findings advance our understanding of the potential roles of both nitrogen and phosphorus in tropical forests.” Once again, this study is specific to one study in Panama. There is a limit to how much these results can be generalized beyond this site, although I do recognize that the authors do a much better job of putting these findings in the broader context when referring to prior work. This was not a cross-site comparison across a range of tropical forest and thus has limited ability to generalize beyond other very similar forest types.

4. I don't find the results all that surprising. We know that within 15 years of forest clearing that LAI cannot be distinguished between secondary and old growth forests. We expect N to be limiting in fast growing early successional trees in the tropics and adding N just gets the trees to the “saturation” point a few years faster. To say this is the full successional gradient, is not quite correct since it is still just early, mid and late succession. I fully agree that this work is of high value and whole heartedly commend the authors for undertaking such a substantial effort to evaluate effects of fertilization. It just really seems to support what other work has already shown – they just haven't shown it in high-profile journals.

5. Line 145 – Phosphorus may not have generated a response in the mature forests of THIS system, but it is also important to recognize that the form, timing, and amount of nutrients added is important, as is the forest site. There is evidence in Costa Rica that P is important. For example, a litter manipulation experiment in Costa Rica found a positive effect of litter addition on forest productivity that was positively correlated with the total P in the added litter (Work by Dr. Wood). This held for some forest sites included, but not all (i.e., low P, secondary versus mature). In contrast, the long-term litter addition experiment in Panama (work by Dr. Sayer) found no effect of litter addition and P didn't seem to be important. I am not sure of the location of this site in Panama versus the litter manipulation experiment of Dr. Sayer, but it seems that these are important considerations that the forest in Panama may not represent other tropical forest site that have more rainfall or other considerations. Further, evaluation of nutrient resorption across a P gradient in Costa Rica found that resorption of P was greater during periods of greater reproductive effort (work by Dr. Tully), and in that same forest, there was a mid-day draw down of available soil P that was linked with stem flow. Perhaps that was passive uptake, but it suggests that at least for other forest that P may be important in mature forests, and that as the authors suggest the effect of P may manifest in components other than woody growth.

Honestly – it would help to know what the typical rate of woody growth is for these forests and whether they have the capacity to grow any faster than they already are. I appreciate that the authors touch on these alternate explanations, as well as the discussion of the community composition. My sentiment holds that if an organism is adapted to low P environment, then its competitive advantage is being able to acquire these components under limiting conditions. If we add tons of fertilizer to plant that is adapted to low nutrients and the plant doesn't grow faster or growth declines, does that mean that it doesn't exist in low nutrient environment and that the system is not limited? or does it really mean the plant itself has limited capacity to respond to more? Mature tropical trees tend to be much slower growing. And there are adaptive strategies for getting rock derived P that may not provide advantageous when tons of it are dumped as fertilizer on the soil. We know from agricultural work and mass production of plants that different species have different nutrient, soil and water requirements. So really, what do we mean when we discuss limitation? Is it really that the organism grows faster when more is added? Or is it that they are adapted to limiting conditions and there are strategies that allow that organism to out compete others under specific limiting conditions that also prevents them from being able to respond to excess. Examples of efficient cycling could be efficient resorption of that element (i.e., Dr. Vitousek, Dr. Killingbeck, among others), changes in allocation belowground, changes in leaf lifespan, changes in phenology or reproductive efforts. While woody growth and aboveground biomass are definitely important for carbon storage, and if that's all we care about, then knowing that mature trees in Panama don't grow faster with N and P is of value. But if the goal is to fully evaluate nutrient limitation, there are other ways that plants can respond and this is just one of them.

6. Fundamentally, when substituting space for time with manipulation experiments, the capacity for adaptation, evolution, etc. is missed, especially when considering the long lifespan of mature tropical trees (as the authors rightly note). How plastic is the ability of these trees to respond to nutrients? Definitely in young, fast-growing trees it makes sense that they would respond, but already mature trees? I don't doubt the results, but I am not convinced by the interpretation, as much as I do appreciate that further discussion is given to these ideas.

7. Regarding the suggestions for managing tropical forest reforestation. Basically, what the authors are saying is that forests reach “saturated” above ground biomass faster when fertilized, which means this effect is a transient boost that goes away in maybe 5 years given that at 15-20 years tropical forests are pretty well developed (i.e., Dr. Chazdon's work). The authors go from 10 years to 30 years, so we don't really know, but I feel like the benefits might be overstated. It isn't that these forests add an additional Megaton of C carbon forever.

8. The entire discussion of forest management doesn't feel appropriate for this paper given that this is not a reforestation study. I think the suggestion that this forest might be limited by some other nutrient – with no example or suggested alternate listed – is indeed interesting, but it doesn't provide us with a way forward for solving this major question in tropical research. And given that focus, I don't think this discussion of management is appropriate in this manuscript. I think it should be a separate commentary or included in a study that is actually studying reforestation and carbon sequestration.

Overall, I think this work is of high value scientifically. However, it failed to provide me with enough convincing evidence for me to say it's time to put the nail in the coffin for P limitation in mature tropical forests, especially when there isn't any evidence of what the “other” limiting factor or element might be and the limited response variables considered. Perhaps this forest is growing as fast as it possibly can given the physiological constraints of mature tropical hardwoods, or potentially the timescale of research isn't long enough to capture increased growth given the slow rate of diameter growth. Further, once a forest is mature and reaches the forest canopy, maybe it is more beneficial to invest excess nutrients elsewhere, for example

- turnover of leaves, or in reproduction. Younger leaves tend to have more nutrients and thus higher rates of photosynthesis and with added nutrients the response might instead be reducing leaf lifespan and increasing turnover, which may not be captured in traditional measures of diameter growth, or that less energy is put into resorption and more into other components. The authors touch on these subjects, but I think other lines of evidence are needed for the scientific community to fully rule out the importance of P in favor of searching for an elusive "other" limiting element. The mystery of what drives mature tropical forest productivity remains, and ultimately, this work didn't move us closer to solving it. Further, while the experimental design is very robust for this site in Panama and certainly a major undertaking, I tend to be very cautious when it comes to generalizing single tropical forest sites too broadly.

Finally, as I mentioned above, I don't think the management discussion works in this paper. The paper is focusing on evaluating nutrient limitation during natural succession, and to then translate this to how we grow trees and where in the tropics is well beyond the scope, especially given the number of people that are actively working on reforestation projects.

(Remarks on code availability)

Version 1:

Reviewer comments:

Reviewer #1

(Remarks to the Author)

This is the third round of reviews that I have conducted on this manuscript, and I believe it is now ready for publication. Compared to the previous revision, I believe the manuscript has improved further. The authors are more concise and specific, and they have toned down the generalisation of their results even further.

They put a lot of work into analysing the available soil analytical data, especially the total phosphorus analyses, and I can accept the choices they made, even if I would have made different ones.

The authors now correctly claim that Agua Salud soil are less weathered than Oxisols, but insist that the soils at Gigante are Oxisols and refer in their rebuttal to the classification made by Ben Turner in 2009. To be clear: I have no doubt that you can find Oxisols in Gigante, but I maintain that a substantial part is also less weathered as was shown in the Koehler et al (2009) paper. Actually, Ben Turner's sampling in 2009 was a direct reaction to the Koehler et al. (2009) paper to prove that he was correct that the soils were Oxisols. And frankly, any soil scientist will know where to sample in the Gigante landscape if you want to find Oxisols. So, in the end the authors decide to make a different choice as I would have made. That does not make them more correct but I can accept this.

Another problem the authors encountered is that, in general, phosphorus analytical data, including total phosphorus, are difficult to compare due to the variety of existing methods. In their response, the authors refer to digestion with nitric acid as the 'gold standard'. Perhaps that is true where they live, but it is not true where I am from. In my understanding, a method that measures 'total phosphorus' tries to measure as much phosphorus as possible in a sample, so if conducted correctly, a method that measures more is probably better for 'total' phosphorus than a method that fails to measure some of the phosphorus present. Nevertheless, I think we can now all agree that the Panama sites are not representative of the part of the Amazon basin with the most weathered soils. The fact that 25% of Amazonian forests have more total phosphorus than the Panama sites is not actually surprising, since a substantial proportion of the region is covered by soils that are less weathered than Oxisols (see Quesada et al., 2009).

I was glad to see that the authors had now given serious thought to the effects of deforestation and slash-and-burn management on nutrient availability. Initially, I was surprised that they did not observe an effect of soil pH, base saturation or total phosphorus (P), but examining the soil chemical data from Gigante (from Ben Turner's sampling mentioned earlier) reveals that this is probably not so surprising, given that the soils in Gigante, even in areas classified as Oxisols, have a relatively high pH (above 5) and high base saturation (above 50% in the top 50 cm and above 80% in the topsoil). Given these initial conditions, it is not surprising that the impact of ash input from a few decades earlier was not measurable against these background values. I think these effects will be much clearer and long-lasting in sites where the original forest has much lower pH, base saturation and total P values. Nevertheless, I still expect that, under such conditions, the nutrient input from ashes following slash-and-burn will have a significant impact on the regrowth of secondary tropical forests and potential nutrient limitations.

In summary, as I mentioned in my earlier review, the study presents new, noteworthy and exciting insights in shifts in nutrient limitation across a secondary succession gradient on soils at the high end of total phosphorus content. I recommend to publish it, as it will make an important contribution to the discussion on nutrient limitation of secondary tropical forests.

(Remarks on code availability)

RESPONSE TO REVIEWER COMMENTS

Reviewer #1 (Remarks to the Author):

I reviewed an earlier version of this manuscript submitted to Nature. Compared to the earlier version there are several things that have improved substantially. Several earlier, somewhat exaggerated claims have disappeared and language usage is now much more concise compared to the earlier version. The authors also better acknowledge and discuss earlier work on N limitation in tropical forests which also improves the clarity on what is new and noteworthy in the current study.

Thanks for reading our draft again and for providing insightful comments to help improve our paper. We appreciate that you recognize that we have improved our manuscript – your feedback has been very valuable.

When considering the critical question how well the soils in Panama represent soils under tropical forests, it was interesting to learn that the authors now acknowledge that both sites (Gigante and Agua Salud) also have less weathered soils than Oxisols. The authors however argue that the total P values are more important than the classification and I agree with that.

We appreciate that the reviewer agrees that total P values are more important than the soil classification, as total P values may vary a lot within one soil order. Interestingly, the total P values in our young Agua Salud soils are lower than the older Gigante soils. For clarification, while the Agua Salud soils are now correctly reported as less weathered than Oxisols, the soils at Gigante are classified as Oxisols – the data can be found here (<https://figshare.com/s/1d048546bfa4efcc9cb6>).

In their answer to my earlier questions regarding the total P values of 600 mg/kg the authors explain that the high number came from the paper by (Mirabello et al., 2013) which apparently overestimated the P in the 'hot' fractions. I looked up this paper and indeed they suggest that silica interference in phosphate detection by molybdate colorimetry may have led to an overestimation. Fortunately, total P was also analyzed in the (Koehler et al., 2009) paper, which was mentioned by the authors in their rebuttal. Koehler et al. (2009) was conducted in the Gigante Experiment and they used pressure digestion in concentrated HNO₃ followed by analysis of the digests using inductively coupled plasma-atomic emission spectrometer and report them in Table 1 of their paper. This total P method does not use colorimetry for analyses and thus does not suffer from potential silicon interference. The numbers reported by Koehler et al, 2009 for the control plots of the Gigante experiment are 550 mg kg⁻¹ in the top 0.05m of mineral soil and still as much as 400 mg kg⁻¹ in the 0.05 – 0.5m depth interval. If we take the lower value of 400 mg kg⁻¹, this would still put the Gigante site in the top 10% values of total P in the Quesada et al., 2010 paper. So even when I accept that the 600ppm reported by Mirabello et al., 2013 were overestimated because of methodological problems, the total P values of Gigante are more comparable to the top 10% of Amazonian forests. And as I pointed out in my review of the earlier version, this was exactly what I would expect of soils developed from basaltic parent material in tectonically active regions such as Panama. However, the real question for this manuscript is whether the total P levels are

so important to extrapolate the result to other parts of the tropics and below I will argue, why I think that is not the case.

The reviewer raises further questions about the different values of total phosphorus reported for our Gigante soils which has prompted us to take a full inventory and critical analysis of differences in the values of soil phosphorus measured at Gigante over the course of the experiment. We have now compiled all total phosphorus data that has been collected for soils at Gigante, which is reported in Extended Data Table 2.

Soils in the top 10-15 cm of control plots and nearby soil pits at Gigante ranged from 303 to 680 mg P kg⁻¹. There could be two sources of inconsistency. First, soils can be highly variable in total phosphorus in central Panama. Second, the four different methods used to measure total phosphorus can result in different values. We previously discounted the value of 680 mg kg⁻¹ reported in Mirabello et al. 2013 because of the issue of the Hedley fractionation method overestimating total phosphorus content when summing all fractions as well as the interference issues discussed above. The reviewer raises the values of 400 to 550 mg kg⁻¹ reported in Koehler et al. 2006. As we write in the Extended Data Table 2, “this soil was measured using pressure digestion with nitric acid followed by analysis with inductively coupled plasma-atomic emission spectrometry⁸⁹ (henceforth pressure digestion). Pressure digestion has an array of operational concerns and safety limitations that can cause errors^{90,91}.” In addition, this point is much higher than the other values, and thus we also discount it. Soils were measured with ignition (550C) and extraction on four occasions; however, as we write in Extended Data Table 2, ignition and extraction is known to underestimate total phosphorus. Finally, on two occasions we measured soils with nitric acid extraction. This is the gold standard estimate of total phosphorus, and we believe provides the best estimate of total phosphorus at Gigante. Thus, we use these values to compare to Agua Salud and the Amazon. Extended Data Table 2 summarizes all of the methods and data.

Now that we have evaluated the data sources and determined the total phosphorus levels at Gigante, we re-evaluate the fractional land area of the Amazon with total phosphorus higher than our sites. We realize that we did not explain how we estimated this in our previous response, and why our estimate (25%+) differs from that of the reviewer (10%). We believe that the latter method considers the fraction of sites, whereas we used the fraction of forested land area across the Amazon with different soil types. To do this, we combined the mean total phosphorus for each soil type for forests across the Amazon Basin (Quesada et al 2010) and the fraction of the Amazon Basin forests that has each soil type (Quesada et al 2011). We find that ~25% of the Amazonian forests have more mean phosphorus than our forests. We conclude that our sites are relatively high in total phosphorus compared to some Amazonian forests, but there is substantial land area in the Amazon with even more total phosphorus than our sites.

We have updated the Extended Data Table 3.

We have updated the soils data for our experimental sites in Extended Data Table 1, updated the text in the manuscript in lines 89 and 195-200, included the data for total phosphorus in Extended data table 2, made open a table with calculations to scale total phosphorus in surface to deep soil layers, and provided the comparison of Amazon soils to ours in Extended Data Table 3.

References

C.A. Quesada, J. Lloyd, M. Schwarz, Patiño, S., T.R. Baker, C. Czimczik, N.M. Fyllas, L. Martinelli, G.B. Nardoto, J. Schmerler, A.J.B. Santos, M.G. Hodnett, R. Herrera, F.J. Luizão, A. Arneith, G. Lloyd, N. Dezzeo, I. Hilke, I. Kuhlmann, M. Raessler, W.A. Brand, H. Geilmann, J.O. Moraes Filho, F.P. Carvalho, R.N. Araujo Filho, J.E. Chaves, O.F. Cruz Junior, T.P. Pimentel, R. Paiva, Variations in chemical and physical properties of Amazon forest soils in relation to their genesis. *Biogeosciences* **7**, 1515–1541 (2010). <https://doi.org/10.5194/bg-7-1515-2010>

Quesada, C. et al. Soils of Amazonia with particular reference to the RAINFOR sites. *Biogeosciences* **8**, 1415–1440 (2011). <https://doi.org/10.5194/bgd-6-3851-2009>

In their rebuttal the authors also write that it may be possible that the younger secondary forests could have a stronger nutrient (P) limitation compared to mature forests due to the rapid relative growth rates of young trees. In the manuscript, theoretical biogeochemical models are also mentioned as the basis for this expectation (l. 44 – l.49). Furthermore, they write counter to expectations from biogeochemical theory, they found little evidence of P limitation in the experiment (L. 122). Although this may be true, I think that the authors miss to appreciate the potential importance of land use history in this case (and in general with secondary forests). A strong P limitation in young secondary forest is unlikely because the secondary forest grows on areas that were previously cleared which probably involved slash-and-burn procedures and potentially agricultural practices. The results of slash and burn is that an important part of the rock derived-nutrients, including P, end up in the soil and the soil pH increases, effects that can last for decades. A higher pH leads to higher P availability and a higher base saturation which means that it is quite unlikely that rock-derived nutrients are limiting in the earlier successional stages. Something that was more or less shown for P by (Wong et al., 2024). To my knowledge, the Walker and Syers model considers natural systems without considering previous land use. However, taking the previous land use into account will go a long way to explain at least some of the findings in your study and I think that at the moment this is not sufficiently done. You discuss the possibility that P limitation is not observed in wood productivity (l 177 and further), you discuss that it is possibly less common than previously thought (L 189 and further), but you never discuss that forest clearing and burning leads to substantial increases in rock-derived nutrients and their availability, which I think is currently missing. You should also not forget that the data from the (Quesada et al., 2010) paper were collected in old-growth forest, where presumable no slash and burn has been taken place in the last centuries. This is presently ignored in the manuscript when you compare your soils with soil of the Amazon Basin, and I really think it is misleading and causes a lot of confusion (at least it took me two reviews to realize what was wrong). I expect that young secondary forests and abandoned agricultural areas will have higher values in total and available nutrients because of former slash-and-burn activities, and that should be highlighted in your discussion. It should not be too difficult to back this up with some studies. BTW, thank you for providing the full soil analysis for the Agua Salud project. Your soil chemical analyses show pH-water values around 5 which is a pH value when soil P is relatively well available. Also, base saturation is regularly above 50% in the top soil which may be the result of ashes from earlier forest clearing and also indicates that these are not very nutrient poor soils.

Your point that total phosphorus can increase following slash and burning of forests is very important. We have added this to the manuscript in lines 50-58 and 204-213

including references to the recent literature. However, our young forests were pastures for the past 30 or 40 years prior to transitioning to forests and did not experience slash and burn during that time since initial deforestation. It is possible that total and available rock derived nutrients may be higher in individual Agua Salud plots than they otherwise would have been in the absence of slash and burn agriculture. Unfortunately, we cannot test this because we do not have pre-deforestation phosphorus or pH values to compare with our recent measurements in the same plots. We now recognize this in the text. However, we can compare our young forests to our old forests with the expectation that phosphorus and pH would be higher in the young forests. The data for total and available phosphorus (Extended data table 1) do not show clear patterns of elevated phosphorus in the young forests relative to our mature forests as might be expected if they were impacted by biomass burning. There was no significant difference in pH or total and resin available phosphorus in the young forests compared to older forests (Extended data Table 1). We also compared the resin available phosphorus in our young plots to the wide range of available phosphorus across dozens of sites in Panama (samples for both young and old forests were analysed with the same methods in the same lab; Condit et al 2013). We found that the available phosphorus levels are relatively low compared to the mature forests (Extended Data Table 1). This adds further evidence that our sites are not particularly phosphorus rich.

As we argue above, regardless of whether or not our Agua Salud forests were impacted by slash and burn agriculture in decades past, we still find it useful to compare these numbers to the Amazon and globally to give context for where our study sits in terms of phosphorus.

We have added an evaluation of the potential for slash and burn agriculture to affect phosphorus in our forests in lines 204-213.

Reference

R. Condit, B.M.J. Engelbrecht, D. Pino, R. Pérez, B.L. Turner, Species distributions in response to individual soil nutrients and seasonal drought across a community of tropical trees. *Proceedings of the National Academy of Sciences* **110**, 5064–5068 (2013). <https://doi.org/10.1073/pnas.1218042110>

I am still not convinced about your implications, in your rebuttal you write that you especially focus on tree planting when suggesting to increase the abundance of N fixing species. I have no doubt that there are some examples of secondary forests with little N fixing trees. However, the (Gei et al., 2018) paper, which was mentioned in the rebuttal, also shows that the majority of secondary forests have a relative basal area of 10% or (much) more. I am not sure whether adding more N-fixing legumes would make a big difference and as far as I know the vast majority of secondary forests is the result of spontaneous regeneration and succession and not from planting.

Thanks for your comments. We recognize the large variation in relative abundance of nitrogen-fixing trees over succession in tropical wet forests; however, note that there are many sites with less than 10% nitrogen-fixers and mean basal area in the Gei et al study is 8.5% at 1 year and 10% at 10 years (from Fig 1c). This suggests 1) nitrogen fixers increase in abundance in the first few years of forest recovery, potentially because of nutrient limitation, competition or dispersal limitation; and 2) forests potentially could support more nitrogen fixers than 10% abundance and/or fix more nitrogen. Thus, increasing fixer abundances either in active reforestation efforts or supplemental planting in managed natural regeneration could make a difference, even in sites with

only 10% fixers. We agree that increased nitrogen fixer abundance may not be enough to relieve all nitrogen limitation and discuss this point in our draft (line 307).

Reference

M. Gei, D.M.A. Rozendaal, L. Poorter, F. Bongers, J. I. Sprent, M.D. Garner, T. Mitchell Aide et al. "Legume abundance along successional and rainfall gradients in Neotropical forests." *Nature ecology & evolution* 2, no. 7 (2018): 1104-1111. <https://doi.org/10.1038/s41559-018-0559-6>

In summary, I think the study presented here definitely gives new, noteworthy and exciting insights into shifts in nitrogen and phosphorus limitations over a secondary succession gradient in soils that are at the higher end of total phosphorus content of soils under (secondary) forests, and as such is worth publishing.

We appreciate your enthusiasm and thank you again for your insightful comments which we believe have substantially improved our manuscript.

I think your claim that your soils are close to the median value for tropical forests soils globally is simply not true and frankly, I don't think it is important for your manuscript. What is important for your message is that in secondary forests, N limitation is decreasing over time, and in my interpretation, it is unlikely to be replaced in the first decades of a secondary forest by P limitation (not even on heavily weathered Oxisols in the Amazon) because of the higher and more available rock-derived nutrient stocks, including soil P, which is a remnant of former slash and burn activities.

We agree with the reviewer's conclusion that our soils are relatively high in phosphorus compared to tropical forests. We revised the sentence, please see lines 193-200. We also appreciate the reviewer's summary of what is the important finding from our study – that nitrogen limitation decreases over time and is unlikely to be replaced by phosphorus limitation, at least in our study area. However, we respectfully disagree that the reason for our lack of phosphorus limitation is because of phosphorus inputs from slash and burn practices in secondary forests. We recognise that this is a possibility; however, our forests have had limited slash and burn treatment decades ago and soil indicators suggest that there was not a substantial influx of phosphorus. Nonetheless, we now acknowledge this possibility in the discussion (lines 204-213).

References

- Gei, M., et al., 2018. Legume abundance along successional and rainfall gradients in Neotropical forests. *Nat. Ecol. Evol.* 2, 1104–1111. <https://doi.org/10.1038/s41559-018-0559-6>
- Koehler, B., et al. 2009. Immediate and long-term nitrogen oxide emissions from tropical forest soils exposed to elevated nitrogen input. *Glob. Change Biol.* 15, 2049–2066. <https://doi.org/10.1111/j.1365-2486.2008.01826.x>
- Mirabello, M.J., et al. 2013. Soil phosphorus responses to chronic nutrient fertilisation and seasonal drought in a humid lowland forest, Panama. *Soil Res.* 51, 215. <https://doi.org/10.1071/sr12188>
- Quesada, C.A., et al., 2010. Variations in chemical and physical properties of Amazon forest soils in relation to their genesis. *Biogeosciences* 7, 1515–1541. <https://doi.org/10.5194/bg-7-1515-2010>
- Wong, M.Y., et al. 2024. Trees adjust nutrient acquisition strategies across tropical forest

secondary succession. *New Phytol.* 243, 132–144. <https://doi.org/10.1111/nph.19812>

Reviewer #2 (Remarks to the Author):

This is the second time I have reviewed this paper (albeit for a different journal). So I am copying out my previous review below since it addresses many of the stock questions asked by NCOMMS.

The authors now present the carbon numbers with appropriate nuance. They have also directly addressed my concern about nitrogen being confounded with age at the beginning. And the methods are now clearly documented, and are sound. I have no additional critiques beyond the two minor grammatical catches below.

Thank you very much for reading our paper and provide constructive comments. We address your new comments below. We addressed all of the comments from the previous review in our last revision, and appreciate that the reviewer recognizes this.

Line 39: “increasing predominance in modern tropical landscapes” would be better than “increasing importance” since the latter is normative

Done

Line 194: in LATIN America? Hawaii is also USA.

Thank you for catching this. We have changed it to “Central and South America” since we prefer to avoid the use of the term “Latin” American due to it’s Eurocentric and colonial origins. Please see the line 241.

We do not respond to the comments from Reviewer 2 below because we addressed them in full in the last revision.

The paper by Tang et al examines impacts of fertilization on a network of tropical forest plots within Panama. They provide supplemental nitrogen and phosphorus to forests that vary in age (~0, 10, 30, and 600 years old) and find that the nitrogen addition greatly augments growth in the young forests only and that – despite standard biogeochemical theory – woody biomass accumulation does not appear to be limited by phosphorus.

The paper is very well written. I have no editorial critiques for clarity, and this never happens when I review a manuscript. As I was reading, multiple questions arose in my mind but they addressed these later in the manuscript. For example, they find that phosphorus is not limiting which is a surprising result, but then provide a very nice discussion to explore potential mechanisms underlying their results. I don’t think the paper’s findings are enough to upend theory, but it certainly suggests that more work is needed to determine the universality of phosphorus limitation in the tropics.

I was concerned that the interaction between nitrogen and forest age was confounded by nitrogen conditions in the soil (i.e., more nitrogen intact forest soils), but they provide

baseline soil fertility conditions in a supplemental information (Extended Data 2) that shows this is not the case. I suggest making that point explicitly within the main manuscript.

I was also concerned that the applied take home from the paper would be that people should apply fertilizer to young regrowing forests to augment growth and carbon removal. But the paper very clearly states that this is not recommended (I agree!) and provide alternative options suggest as prioritizing regrowth in places with N deposition.

However, the largest issue is reproducibility. The methods are non-existent. The paragraph located lines 68-78 seems written to make the experiment sound larger than it actually is. They say there are 76 plots, but I think this includes both the treatment and the control so actually the number of experimental units is smaller. Sample size per treatment appears to be restricted to the figure legends. They say that these cover 88,843 trees, but those are not independent replicates. They report that they monitored these plots for up to 21 years and then mention in the parentheses that secondary forests have been monitored for four, but in fact the main results of the paper come from those secondary forests. How large are the plots? They say they cover 16 km², but what is the range of sizes for the individual plots? Where are they located (i.e., can the geolocations be made available)? How much fertilizer did they add? What dosage, what frequency? How did they quantify biomass (i.e., which allometric equations were used)? This is all easily remedied by including a clear methods section – in the supplement if no space is available in the main text.

My only other critique is that their carbon numbers require some nuance. I appreciate that they estimated mitigation potential at a cost-constrained rather than unconstrained level. However, their number remains quite theoretical. It is highly unlikely/impossible to achieve 1.1 GtCO₂/yr because there are only so many locations with N deposition or higher fertility soils and only so much N that can be provided by nitrogen fixing species. Adding fertilizer to relieve nutrient limitations would produce upstream and in-field emissions that would offset the climate benefit of enhanced forest growth. I suggest the co-authors nuance that result more appropriately. For example, in the abstract instead of “could sequester up to an additional 1.1 GtCO₂/yr” it might say “N limitations prevent up to 1.1 GtCO₂/yr”. In other words, that 1.1 GtCO₂/yr shouldn't be presented as an opportunity.

Optional suggestions:

(1) Table 1 could be supplemental

(2) Figure 1 should be checked for color blind friendliness. And would a log-scale for the y-axis better illustrate the differences among treatments (as an alternative to the inset approach).

Reviewer #2 (Remarks on code availability):

Code and metadata are appropriate.

Reviewer #3 (Remarks to the Author):

This study evaluates the effects of fertilization on aboveground biomass accumulation in early, mid and late successional forests in Panama. The work itself is incredibly ambitious, and the results are very robust and of interest. This study was a major undertaking. The manuscript is also beautifully written, and the authors certainly present some intriguing ideas. I read an earlier draft of this manuscript, and there are substantial improvements. I appreciate the broader acknowledgement prior work, and the more developed discussion of alternate explanations for the lack of response to N and P fertilization in mature forests. That said, many of my prior reservations hold.

Thank you very much for reading our draft and providing so many thoughtful comments. We appreciate that you recognize the ambitiousness and robustness of our study. We also value that you see our improvements following your helpful comments from a previous review. We address your reservations below.

1. Abstract: I appreciate that in the discussion, the authors compare this work to forests in the Amazon, but I am still extremely hesitant about generalizing findings from a single forest site in Panama to the tropics as a whole, especially given the wide range of tropical forest types (i.e., seasonal, evergreen, flooded, etc) and that the neotropics may not reflect what we would see in Asian forests or other less studied systems.

We take your point and have now specified that we are discussing results from our forests in the abstract and add a caveat to the line about scaling up. We write (new text in bold): “**Central American landscape**” and “Nutrient limitation of aboveground biomass accumulation shifts over succession **in our forests**” in the abstract (lines 27-29). Finally, we modified the last sentence of the abstract to read, “Our findings suggest that, **if nitrogen limitation is widespread in young recovering tropical forests**, nitrogen limitation **could** prevent up to 0.84 Gt CO₂ uptake by recovering tropical forests each year” (lines 33-35). We further reduce generalization in other parts of the manuscript besides the abstract, including lines 198-200, and discuss the possibility of heterogeneity in nutrient limitation in young forests (164-167).

2. Regarding the following statement –

“To date, we are aware of no direct ecosystem experiment that investigates how nutrient limitation on stand-scale aboveground carbon dynamics evolves over the course of a complete tropical forest secondary succession gradient, from recently abandoned pastures to mature forests.”

And this matters why? Being the first, or only, in and of itself isn't that interesting, but something that helps fill a specific knowledge gap, is. Please elaborate on why we as a scientific community should care about having this additional detail, and if the experiment just supports what is already generally accepted amongst tropical forest ecologists (i.e., early successional forests are N limited), is this really transforming our understanding? Definitely this is a really well-designed study, and the findings are very robust, but it just supports what others have shown - although maybe not as well.

We now understand your point, and have deleted this sentence and rearranged the introduction to make a clearer argument identifying the knowledge gap and how we address it in lines 53-82.

The authors suggest that some element other than P is what is really limiting mature tropical forests, but what do they think it is?. Also -- why would we expect mature trees to respond in the same way as fast growing early successional trees that are working to access the forest canopy?

Thanks for your suggestion. We have added a sentence that mature forest carbon sink could be limited by other resources like potassium (Wright et al., 2011; Manu et al. 2022), calcium (Bauters et al., 2022) or water (Tao et al., 2022) (lines 256-257).

You have a good point about the possibility that the mature forest trees may not be able to respond to nutrients in the same way that young forest trees can. For example, the mature forest trees may be adapted to lower phosphorus conditions and may not be able to respond to added phosphorus. We have added some sentences about this in lines 224-238, 240-249.

References

S.J. Wright, J.B. Yavitt, N. Wurzbarger, B.L. Turner, E.V.J. Tanner, E.J. Sayer, L.S. Santiago, M. Kaspari, L.O. Hedin, K.E. Harms, M.N. Garcia, M.D. Corre, Potassium, phosphorus, or nitrogen limit root allocation, tree growth, or litter production in a lowland tropical forest. *Ecology* **92**, 1616–1625 (2011). <https://doi.org/10.1890/10-1558.1>

R. Manu, M.D. Corre, A. Aleeje, M.J.G. Mwanjalolo, F. Babweteera, E. Veldkamp, O. van Straaten, Responses of tree growth and biomass production to nutrient addition in a semi-deciduous tropical forest in Africa. *Ecology* **103**, e3659 (2022). <https://doi.org/10.1002/ecy.3659>

M. Bauters, I.A. Janssens, D. Wasner, S. Doetterl, P. Vermeir, M. Griepentrog, T.W. Drake, J. Six, M. Barthel, S. Baumgartner, K. Van Oost, I.A. Makelele, C. Ewango, K. Verheyen, P. Boeckx. Increasing calcium scarcity along Afrotropical forest succession. *Nature Ecology & Evolution* **6**, 1122–1131 (2022).

S. Tao, J. Chave, P.L. Frison, T.L. Toan, P. Ciais, J. Fang, J.P. Wigneron, M. Santoro, H. Yang, X. Liu, N. Labriere, Increasing and widespread vulnerability of intact tropical rainforests to repeated droughts. *Proceedings of the National Academy of Sciences* **119**, e2116626119 (2022). <https://doi.org/10.1073/pnas.2116626119>

3. The following statement “These findings advance our understanding of the potential roles of both nitrogen and phosphorus in tropical forests.” Once again, this study is specific to one study in Panama. There is a limit to how much these results can be generalized beyond this site, although I do recognize that the authors do a much better job of putting these findings in the broader context when referring to prior work. This was not a cross-site comparison across a range of tropical forest and thus has limited ability to generalize beyond other very similar forest types.

We agree our study is specific to one area and therefore may not reflect limitation in other tropical forests. Nonetheless, we still believe that the findings may have relevance to other tropical forests, especially because the results follow what has previously been found in several studies (as you mention above). We have deleted this sentence and replaced it with, “Although they derive from one site, these findings advance our

understanding of the role that nitrogen and phosphorus can have in tropical forests” (lines 153-154). We also added a sentence in lines 202-203, “Future work should examine the degree to which these patterns are consistent in other tropical forests, including African and Asian tropical forests.”

4. I don't find the results all that surprising. We know that within 15 years of forest clearing that LAI cannot be distinguished between secondary and old growth forests. We expect N to be limiting in fast growing early successional trees in the tropics and adding N just gets the trees to the “saturation” point a few years faster.

Although there is some experimental and observational evidence that suggests nitrogen limits young forests, a meta-analysis with all field-based fertilization experiments found that either nitrogen or phosphorus can be limiting young tropical forests (Wright, 2019). Furthermore, comparison of results from different experiments in different sites may suffer from confounding factors that hinder inference. Thus, nitrogen limitation in our young forests was not a forgone conclusion, and the shift in nutrient limitation within one forest area is novel. We have substantially reorganized our introduction to more clearly highlight the novelty of our experiment. Please see lines 60-77.

Reference

S.J. Wright, Plant responses to nutrient addition experiments conducted in tropical forests. *Ecological Monographs* **89**, e01382 (2019). <https://doi.org/10.1002/ecm.1382>

To say this is the full successional gradient, is not quite correct since it is still just early, mid and late succession. I fully agree that this work is of high value and whole heartedly commend the authors for undertaking such a substantial effort to evaluate effects of fertilization. It just really seems to support what other work has already shown – they just haven't shown it in high-profile journals.

We have removed the word “complete” in lines 27, 85, 189.

5. Line 145 – Phosphorus may not have generated a response in the mature forests of THIS system, but it is also important to recognize that the form, timing, and amount of nutrients added is important, as is the forest site. There is evidence in Costa Rica that P is important. For example, a litter manipulation experiment in Costa Rica found a positive effect of litter addition on forest productivity that was positively correlated with the total P in the added litter (Work by Dr. Wood). This held for some forest sites included, but not all (i.e., low P, secondary versus mature). In contrast, the long-term litter addition experiment in Panama (work by Dr. Sayer) found no effect of litter addition and P didn't seem to be important. I am not sure of the location of this site in Panama versus the litter manipulation experiment of Dr. Sayer, but it seems that these are important considerations that the forest in Panama may not represent other tropical forest site that have more rainfall or other considerations. Further, evaluation of nutrient resorption across a P gradient in Costa Rica found that resorption of P was greater during periods of greater reproductive effort (work by Dr. Tully), and in that same forest, there was a mid-day draw down of available soil P that was linked with stem flow. Perhaps that was passive uptake, but it suggests that at least for other forest that P may be important in mature forests, and that as the authors suggest the effect of P may manifest in components other than woody growth.

We do not intend to argue that phosphorus is not important in other tropical forests or even in our sites (see, for example, Wong et al 2024 showing that phosphorus influences some nutrient acquisition strategies in our experiment). In fact, we see our study as contributing to the increasing evidence that nutrient limitation in tropical forests is heterogeneous and more complex than just influencing aboveground woody growth (see lines 159-170 and 231-238). We agree with the reviewer that there are many factors that could determine whether a response to phosphorus is generated including site, and that other components of trees may be affected by phosphorus while woody growth is not (lines 215-217). For reference, the litter manipulation experiment that Dr. Sayer maintains is adjacent to our mature forest fertilization experiment on the same Gigante peninsula. At that experiment, the current hypothesis is that litter removal induced nitrogen rather than phosphorus limitation because litter removal carried away 25 times more nitrogen than phosphorus (Tanner et al 2024, *The Gigante Litter Manipulation Experiment*, from *The First 100 Years of Research on Barro Colorado: Plant and Ecosystem Science*, vol. 2). We have expanded our discussion of potential heterogeneity in nutrient limitation in tropical forests (lines 231-238) and have explicitly acknowledged that some tropical forests may be phosphorus limited by adding the sentence that “Nonetheless, evidence suggests that phosphorus may limit some tropical forests” (lines 255-260).

Reference

C_Muller-Landau, Helene, and S. Joseph Wright. "The First 100 Years of Research on Barro Colorado: Plant and Ecosystem Science (Volumes 1 and 2)." (2024).

Honestly – it would help to know what the typical rate of woody growth is for these forests and whether they have the capacity to grow any faster than they already are. I appreciate that the authors touch on these alternate explanations, as well as the discussion of the community composition. My sentiment holds that if an organism is adapted to low P environment, then its competitive advantage is being able to acquire these components under limiting conditions. If we add tons of fertilizer to plant that is adapted to low nutrients and the plant doesn't grow faster or growth declines, does that mean that it doesn't exist in low nutrient environment and that the system is not limited? or does it really mean the plant itself has limited capacity to respond to more? Mature tropical trees tend to be much slower growing. And there are adaptive strategies for getting rock derived P that may not provide advantageous when tons of it are dumped as fertilizer on the soil. We know from agricultural work and mass production of plants that different species have different nutrient, soil and water requirements. So really, what do we mean when we discuss limitation? Is it really that the organism grows faster when more is added? Or is it that they are adapted to limiting conditions and there are strategies that allow that organism to out compete others under specific limiting conditions that also prevents them from being able to respond to excess. Examples of efficient cycling could be efficient resorption of that element (i.e., Dr. Vitousek, Dr. Killingbeck, among others), changes in allocation belowground, changes in leaf lifespan, changes in phenology or reproductive efforts. While woody growth and aboveground biomass are definitely important for carbon storage, and if that's all we care about, then

knowing that mature trees in Panama don't grow faster with N and P is of value. But if the goal is to fully evaluate nutrient limitation, there are other ways that plants can respond and this is just one of them.

You raise a good question about whether these forests have the capacity to grow more than they currently do, and thus if we expect them to be able to respond to resource additions. We believe that they do have this capacity, although we agree with the reviewer that we may not be able to observe a response to phosphorus in woody growth because of mismatches in the timescale of our experiment and species turnover. Aboveground woody productivity (AWP) in our mature forest plots ($5.3 \text{ Mg ha}^{-1} \text{ year}^{-1}$) is at the lower end for forests in Panama. There is high small-scale variability in AWP at the nearby 50 ha plot at BCI ranges, where AWP ranges from from 6.4 to $9.2 \text{ Mg ha}^{-1} \text{ year}^{-1}$ across a small-scale topographic gradient within the 50 ha (Piponiot et al 2024)⁵⁰. Furthermore, there is a two-fold range across 39 mostly 1 ha plots located across the Isthmus of Panama, ranging from 3 to $7 \text{ Mg ha}^{-1} \text{ yr}^{-1}$ (Muller-Landau et al. 2024)⁴⁹. Thus, nearby forests can grow up to twice as fast as our forests. We now discuss this on lines 228-231.

We agree with you that the timescales at which a response might occur could exceed that of our experiment because the current trees may be adapted to low nutrient conditions and therefore do not have capacity to respond to phosphorus addition. Over longer timescales, response to phosphorus may occur through changes in species composition from low to high phosphorus adapted species. We have added a paragraph discussing this on lines 224-235. We also agree that trees may adjust nutrient acquisition and use strategies which could make them more efficient in limiting conditions. In fact, we have found that root phosphatase activity increases in the 30-year-old and mature forests and is suppressed by phosphorus in our experiment (Wong et al 2024)³⁴ (lines 244-246). We mention responses of other tissues that have been observed in our mature forests, like increased reproduction (Fortier & Wright, 2021)⁴⁸, in lines 216.

References

C. Piponiot, R. Condit, S.P. Hubbell, R. Pérez, S. Lao, S. Aguilar, H.C. Muller-Landau, Woody Biomass Stocks and Fluxes in the Barro Colorado Island 50-ha Plot. Smithsonian Institution Scholarly Press. Chapter (2024). <https://doi.org/10.5479/si.26882404>

H. Muller-Landau, J. Wright, Looking Forward to the Next 100 Years of Plant and Ecosystem Science at Barro Colorado. Smithsonian Institution Scholarly Press. Chapter (2024). <https://doi.org/10.5479/si.26882656>

M.Y. Wong, N. Wurzbarger, J.S. Hall, S.J. Wright, W. Tang, L.O. Hedin, K. Saltonstall, M. van Breugel, S.A. Batterman, Trees adjust nutrient acquisition strategies across tropical forest secondary succession. *New Phytologist* (2024). <https://doi.org/10.1111/nph.19812>

R. Fortier, S.J. Wright, Nutrient limitation of plant reproduction in a tropical moist forest. *Ecology* **102**, e03469 (2021). <https://doi.org/10.1002/ecy.3469>

6. Fundamentally, when substituting space for time with manipulation experiments, the

capacity for adaptation, evolution, etc. is missed, especially when considering the long lifespan of mature tropical trees (as the authors rightly note). How plastic is the ability of these trees to respond to nutrients? Definitely in young, fast-growing trees it makes sense that they would respond, but already mature trees? I don't doubt the results, but I am not convinced by the interpretation, as much as I do appreciate that further discussion is given to these ideas.

The plasticity question is very important and we appreciate the opportunity to add discussion of flexibility in lines 224-235.

We do have some evidence that the nutrient strategies in our experimental forests are plastic, including root phosphatase activity and mycorrhizal colonization (Wong et al 2024), and this plasticity occurs in all forest ages. However, it is a good question whether this flexibility in nutrient strategies translates to an ability to be plastic in growth rates over short time periods.

Reference:

M.Y. Wong, N. Wurzbarger, J.S. Hall, S.J. Wright, W. Tang, L.O. Hedin, K. Saltonstall, M. van Breugel, S.A. Batterman, Trees adjust nutrient acquisition strategies across tropical forest secondary succession. *New Phytologist* (2024). <https://doi.org/10.1111/nph.19812>

7. Regarding the suggestions for managing tropical forest reforestation. Basically, what the authors are saying is that forests reach "saturated" above ground biomass faster when fertilized, which means this effect is a transient boost that goes away in maybe 5 years given that at 15-20 years tropical forests are pretty well developed (i.e., Dr. Chazdon's work). The authors go from 10 years to 30 years, so we don't really know, but I feel like the benefits might be overstated. It isn't that these forests add an additional Megaton of C carbon forever.

Yes, you are correct that we are finding that nutrients boost forest recovery rates and that the carbon sequestration boost is temporary. We recognize that the rate of forest carbon sequestration due to alleviation of nutrient limitation for aging forests declines over time and that the land area available for reforestation theoretically would also eventually decline. However, this initial boost is important and valuable because natural climate solutions that allow us to sequester more carbon quickly albeit temporarily will buy us time to switch to slower-to-scale solutions that sequester carbon for the long-term. We focus on the next 30 years because this time period is critical for our carbon trajectories and amount of warming.

We have now added nuance to our estimate of how the maximum potential carbon sequestration could be prevented by nitrogen limitation in the future when reforestation is valued as a natural climate solution (e.g., \$100 tCO₂⁻¹). Note that our estimate is an upper bound as noted by the word maximum, since our scaling assumes that all newly regenerating tropical forests have the same type and strength of nutrient limitation as our site, despite the evidence that there could be heterogeneity in the strength and type of limitation in young forests.

We also add nuance to our estimate by considering the aging forest and declining carbon sequestration rate with age. We fine-tune our carbon multiplier by using our data at 4, 14 and 30 years to estimate the carbon multiplier at 10, 20 and 30 years. The 10-year increments align with the forest ages used in model projections from Busch et al

2019 and therefore allows for more precise scaling. We calculate the additional carbon sequestered at 10, 20 and 30 years, assuming a constant rate from 0-4 years and linear declines between 4 and 14 years and 14 and 30 years. We then track the proportion of land area in each decade of forests that are 0-10, 10-20 and 20-30 years old by considering the land area reforested in each decade. The addition of this nuanced method reduces the projection of carbon sequestration prevented by nitrogen limitation, as the reviewer implies, from a maximum of 1.1 Gt CO₂ year⁻¹ to a maximum of 0.84 Gt CO₂ year⁻¹ in 2040-2050. The mean is 0.69 Gt CO₂ year⁻¹ across the three decades. Despite this reduction, the amount of reforestation prevented by nutrient limitation is still substantial because the majority of land newly reforested in each time period is in forests that are less than 10 years old, when the carbon multiplier is highest. From 2020-2030, 100% of projected reforested areas would have forests that are 0-10 years old. From 2030-2040, 53% of projected reforested areas would have forests that are 0-10 years old and 47% would have forests that are 10-20 years old. From 2040-2050, 38% of projected reforested areas would have forests that are 0-10 years old, 33% would have forests that are 10-20 years old and 29% would have forests that are 20-30 years old.

We have expanded our discussion of methods and caveats for this projection in the Supplemental Information 2 and in lines 295-302, which we hope adds clarity.

Reference

J. Busch, J. Engelmann, S.C. Cook-Patton, B.W. Griscom, T. Kroeger, H. Possingham, P. Shyamsundar, Potential for low-cost carbon dioxide removal through tropical reforestation. *Nature Climate Change* 9, 463–466 (2019). <https://doi.org/10.1038/s41558-019-0485-x>

8. The entire discussion of forest management doesn't feel appropriate for this paper given that this is not a reforestation study. I think the suggestion that this forest might be limited by some other nutrient – with no example or suggested alternate listed – is indeed interesting, but it doesn't provide us with a way forward for solving this major question in tropical research. And given that focus, I don't think this discussion of management is appropriate in this manuscript. I think it should be a separate commentary or included in a study that is actually studying reforestation and carbon sequestration.

We appreciate the reviewer's perspective but respectfully disagree that we should cut our management recommendations from the manuscript. Our projection of how much additional carbon could be sequestered because of management practices that alleviate nutrient limitation hinge on all regrowing tropical forests being limited by nitrogen – a substantial uncertainty that we now acknowledge in the main text. Nonetheless, we think it important to provide a maximum bound for how much nitrogen limitation on carbon sequestration there may be – and strategies for alleviating that limitation – in recovering forests reforested as natural climate solutions.

Furthermore, we disagree with the reviewer's argument that we should not include management recommendations because we do not resolve what limits older tropical forests. The type of nutrient limitation in older forests does not influence our carbon sequestration estimates because, as the reviewer acknowledges, several other lines of evidence along with our study provide substantial evidence that nitrogen limits young forests, and we remain nutrient limitation type-neutral in our projections for older forests, assuming no nutrient limitation as forests age (as per our experimental results), which is conservative. We recognize that our experimental results reported here were not able to determine whether a different nutrient besides nitrogen or phosphorus limited older

forests, but any limitation by a different nutrient would provide an opportunity for the sequestration of even more carbon if forests were managed to alleviate that nutrient limitation. Thus, we believe that our carbon projections and management recommendations are valuable, despite the fact that we did not resolve what limits mature tropical forests.

We also believe that our recommendations for management are valuable even though “this is not a reforestation study” because we indeed designed it as a reforestation study – one of natural regeneration. Our main goals were to understand how nutrient limitation patterns shift along forest succession and provide recommendations for policymakers to increase carbon storage potential of reforestation projects as a natural climate solution. We have now updated our introduction to specifically mention tropical reforestation as a natural climate solution and how our study could benefit the implementation and management of tropical reforestation (lines 40-44, 94-98). Indeed, understanding controls of natural regeneration rates is critical for reforestation practices because it is the main method of reforestation used in the tropics today (Gao et al 2025), is pledged to be used in over a third of total area to be reforested under the Bonn Challenge (Lewis et al 2019), and could be used in much more land area in the future, including in many of the areas pledged for reforestation with plantations under the Bonn Challenge (Chazdon and Guariguata 2016, Crouzeilles et al 2017, Lewis et al 2019, Williams et al 2024). We are conservative in our carbon projections because we only consider additional land reforested when using reforestation as a natural climate solution (that resulting from a carbon price), but reforestation that occurs in the future under business-as-usual scenarios could also sequester more carbon, which would also be additional, if it is done intentionally with our management recommendations.

Resolving the role of nutrients in limiting natural forest regeneration is an important knowledge gap. Previous studies of controls of carbon sequestration rates over secondary succession in American tropical forests have failed to identify nutrients as a key factor (e.g., Poorter et al 2016). Our findings about nitrogen’s effect can inform reforestation efforts and estimates of the carbon sequestration potential of tropical reforestation. Thus, we find it relevant and important to raise the broader implications in our manuscript. *Nature Communications* appeals to a broad and diverse audience, and thus our study – if published there – has the potential to reach, amongst others, the restoration and natural climate solutions communities, including those actively managing restoration and those making projections of future carbon sequestration potential.

Reference:

X. Gao, P.B. Reich, J.R. Vincent, M.E. Fagan, R.L. Chazdon, S. Fritz, D. Schepaschenko, M.D. Potts, M.C. Hansen, M. Jung, P.H.S. Brancalion, M. Uriarte, T.F. Keenan, T.W. Crowther, R.O. Dubayah, M. Lesiv, S. Liang, D. Wang. The importance of distinguishing between natural and managed tree cover gains in the moist tropics. *Nature Communications* **16**, 6092 (2025). <https://doi.org/10.1038/s41467-025-59196-1>

S. L. Lewis, C. E. Wheeler, E. T. Mitchard, A. Koch. Regenerate natural forests to store carbon. *Nature* **568**, 25–28 (2019). <https://doi.org/10.1038/d41586-019-01026-8>

R. L. Chazdon, & M. R. Guariguata, Natural regeneration as a tool for large- scale forest restoration in the tropics: prospects and challenges. *Biotropica* **48**, 716-730 (2016). <https://doi.org/10.1111/btp.12381>

R. Crouzeilles, M.S. Ferreira, R.L. Chazdon, D.B. Lindenmayer, J.B.B. Sansevero, L. Monteiro, A. Iribarrem, A.E. Latawiec, B.B.N. Stassburg. Ecological restoration success is higher for natural

regeneration than for active restoration in tropical forests. *Science advances* **3**, e1701345 (2017). [https://DOI: 10.1126/sciadv.1701345](https://doi.org/10.1126/sciadv.1701345)

B. A. Williams, H.L. Beyer, M.E. Fagan, R.L. Chazdon, M. Schmoeller, S. Sprenkle-Hyppolite, B.W. Griscom, J.E.M. Watson, A.M. Tedesco, M. Gonzalez-Roglich, G.A. Daldegan, B. Bodin, D. Celentano, S.J. Wilson, J.R. Rhodes, N.S. Alexandre, D.H. Kim, D. Bastos, R. Crouzeilles, Global potential for natural regeneration in deforested tropical regions. *Nature* **636**, 131–137 (2024). <https://doi.org/10.1038/s41586-024-08106-4>

L. Poorter, F. Bongers, T.M. Aide, A.M. Almeyda Zambrano, P. Balvanera, J.M. Becknell, V. Boukili, P.H.S. Brancalion, E.N. Broadbent, R.L. Chazdon, D. Craven, J.S. de Almeida-Cortez, G.A.L. Cabral, B.H.J. de Jong, J.S. Denslow, D.H. Dent, S.J. DeWalt, J.M. Dupuy, S.M. Durán, M.M. Espírito-Santo, M.C. Fandino, R.G. César, J.S. Hall, J.L. Hernandez-Stefanoni, C.C. Jakovac, A.B. Junqueira, D. Kennard, S.G. Letcher, J.-C. Licona, M. Lohbeck, E. Marín-Spiotta, M. Martínez-Ramos, P. Massoca, J.A. Meave, R. Mesquita, F. Mora, R. Muñoz, R. Muscarella, Y.R.F. Nunes, S. Ochoa-Gaona, A.A. de Oliveira, E. Orihuela-Belmonte, M. Peña-Claros, E.A. Pérez-García, D. Piotta, J.S. Powers, J. Rodríguez-Velázquez, I.E. Romero-Pérez, J. Ruíz, J.G. Saldarriaga, A. Sanchez-Azofeifa, N.B. Schwartz, M.K. Steininger, N.G. Swenson, M. Toledo, M. Uriarte, M. van Breugel, H. van der Wal, M.D.M. Veloso, H.F.M. Vester, A. Vicentini, I.C.G. Vieira, T.V. Bentos, G.B. Williamson, D.M.A. Rozendaal, Biomass resilience of Neotropical secondary forests. *Nature* **530**, 211–214 (2016). <https://doi.org/10.1038/nature16512>

Overall, I think this work is of high value scientifically. However, it failed to provide me with enough convincing evidence for me to say it's time to put the nail in the coffin for P limitation in mature tropical forests, especially when there isn't any evidence of what the "other" limiting factor or element might be and the limited response variables considered. Perhaps this forest is growing as fast as it possibly can given the physiological constraints of mature tropical hardwoods, or potentially the timescale of research isn't long enough to capture increased growth given the slow rate of diameter growth. Further, once a forest is mature and reaches the forest canopy, maybe it is more beneficial to invest excess nutrients elsewhere, for example - turnover of leaves, or in reproduction. Younger leaves tend to have more nutrients and thus higher rates of photosynthesis and with added nutrients the response might instead be reducing leaf lifespan and increasing turnover, which may not be captured in traditional measures of diameter growth, or that less energy is put into resorption and more into other components. The authors touch on these subjects, but I think other lines of evidence are needed for the scientific community to fully rule out the importance of P in favor of searching for an elusive "other" limiting element. The mystery of what drives mature tropical forest productivity remains, and ultimately, this work didn't move us closer to solving it. Further, while the experimental design is very robust for this site in Panama and certainly a major undertaking, I tend to be very cautious when it comes to generalizing single tropical forest sites too broadly.

Finally, as I mentioned above, I don't think the management discussion works in this paper. The paper is focusing on evaluating nutrient limitation during natural succession, and to then translate this to how we grow trees and where in the tropics is well beyond the scope, especially given the number of people that are actively working on reforestation projects.

Thank you again for reading our draft and provide so many constructive comments. We very much appreciate that you recognize that our study has high scientific value. We

believe we have vastly improved our manuscript in response to your feedback. We summarize our responses to your conclusions below.

Our goal was not to “put the nail in the coffin for P limitation.” It was to examine limitation at our site over a short time scale, recognizing many caveats that may prevent us from determining whether there is ultimate nutrient limitation on our site (discussed across lines 204-261). In fact, we believe that the possibility of lack of phosphorus limitation at our site adds to the growing body of evidence that nutrient limitation may be much more heterogeneous in the tropics than previously assumed, including that some sites are limited by phosphorus, as some studies show. We discuss this in the paragraph of lines 204-261, with new text on lines 204-213, 224-235, 245-260. We have added discussion that higher growth of our forests seems biophysically possible since nearby forests have productivity up to twice that of our forests (lines 228-233), although we agree with the reviewer that the timescale of our experiments may not be able to capture turnover in tree community, limiting our ability to completely out-rule limitation, especially in the older forests (more completely discussed in lines 233-235). We agree with the reviewer that it may be beneficial for trees in older forests to allocate carbon to other tissues rather than wood, and we discuss this in lines 216-217. We have taken the reviewer’s advice and have provided even more caution in generalizing our study to other tropical sites broadly, including on lines 27-28, 34-36, 94-96, 200-204, 274-276. Finally, we appreciate the reviewer’s perspective, but respectfully disagree and believe we should include the management implications for multiple reasons outlined in our response to comment #8.

REVIEWERS' COMMENTS

Reviewer #1 (Remarks to the Author):

This is the third round of reviews that I have conducted on this manuscript, and I believe it is now ready for publication. Compared to the previous revision, I believe the manuscript has improved further. The authors are more concise and specific, and they have toned down the generalisation of their results even further.

They put a lot of work into analysing the available soil analytical data, especially the total phosphorus analyses, and I can accept the choices they made, even if I would have made different ones.

The authors now correctly claim that Agua Salud soil are less weathered than Oxisols, but insist that the soils at Gigante are Oxisols and refer in their rebuttal to the classification made by Ben Turner in 2009. To be clear: I have no doubt that you can find Oxisols in Gigante, but I maintain that a substantial part is also less weathered as was shown in the Koehler et al (2009) paper. Actually, Ben Turner's sampling in 2009 was a direct reaction to the Koehler et al. (2009) paper to prove that he was correct that the soils were Oxisols. And frankly, any soil scientist will know where to sample in the Gigante landscape if you want to find Oxisols. So, in the end the authors decide to make a different choice as I would have made. That does not make them more correct but I can accept this.

Another problem the authors encountered is that, in general, phosphorus analytical data, including total phosphorus, are difficult to compare due to the variety of existing methods. In their response, the authors refer to digestion with nitric acid as the 'gold standard'. Perhaps that is true where they live, but it is not true where I am from. In my understanding, a method that measures 'total phosphorus' tries to measure as much phosphorus as possible in a sample, so if conducted correctly, a method that measures more is probably better for 'total' phosphorus than a method that fails to measure some of the phosphorus present. Nevertheless, I think we can now all agree that the Panama sites are not representative of the part of the Amazon basin with the most weathered soils. The fact that 25% of Amazonian forests have more total phosphorus than the Panama sites is not actually surprising, since a substantial proportion of the region is covered by soils that are less weathered than Oxisols (see Quesada et al., 2009).

I was glad to see that the authors had now given serious thought to the effects of deforestation and slash-and-burn management on nutrient availability. Initially, I was surprised that they did not observe an effect of soil pH, base saturation or total phosphorus (P), but examining the soil chemical data from Gigante (from Ben Turner's sampling mentioned earlier) reveals that this is probably not so surprising,

given that the soils in Gigante, even in areas classified as Oxisols, have a relatively high pH (above 5) and high base saturation (above 50% in the top 50 cm and above 80% in the topsoil). Given these initial conditions, it is not surprising that the impact of ash input from a few decades earlier was not measurable against these background values. I think these effects will be much clearer and long-lasting in sites where the original forest has much lower pH, base saturation and total P values. Nevertheless, I still expect that, under such conditions, the nutrient input from ashes following slash-and-burn will have a significant impact on the regrowth of secondary tropical forests and potential nutrient limitations.

In summary, as I mentioned in my earlier review, the study presents new, noteworthy and exciting insights in shifts in nutrient limitation across a secondary succession gradient on soils at the high end of total phosphorus content. I recommend to publish it, as it will make an important contribution to the discussion on nutrient limitation of secondary tropical forests.

Response:

Thanks again for sending our draft back to the same reviewers, and we appreciate the reviewer's feedback in vastly improving our manuscript in previous revisions.